# Self-Diagnosing Neural Networks: A Causal Framework for Real-Time Anomaly Detection in Training Dynamics

## Abstract

Monitoring neural network training via scalar curves can obscure early indicators of failure in high-dimensional optimization dynamics. This work studies a framework that treats training as a spatiotemporal signal, converting sequences of internal activations and gradients into internal-state videos. A Dynamics Masked Autoencoder (Dynamics-MAE) is pretrained on these videos to learn dynamics-aware representations, and a Temporal Vision Diagnostician (TeViD) with an evidential classification head is then fine-tuned for streaming, open-set diagnosis of training runs under a past-only constraint. Given a short window of internal-state frames, TeViD predicts diagnostic labels (Healthy, Overfitting, Instability, Catastrophic Forgetting, Concept Bias) and can abstain via an *Unknown* category. Evaluation uses time-to-detect, event-time AUPRC, risk–coverage analysis, and a simple decision-theoretic cost model. On $> 500$ held-out runs from a curated data-factory benchmark with factorially held-out architectures, datasets, optimizers, and anomaly types, the method attains an event-time AUPRC of $0.96 \pm 0.01$ and issues alerts a median of $6.2$ epochs earlier than scalar rule-based baselines at a fixed $5\%$ false-alarm rate, suggesting that internal-state video diagnosis can serve as a useful training-time signal alongside existing Machine Learning Operations (MLOps) tools on this benchmark.

## 1 Introduction

Training of modern neural networks is a high-dimensional process with many failure modes—overfitting, optimization instability, data-pipeline issues, catastrophic forgetting, and spurious shortcut reliance. In large-scale settings, a single failed run can waste substantial compute and delay downstream work. Despite mature deployment-time monitoring, experiment tracking, and data-quality checks in Machine Learning Operations (MLOps) (Berberi et al., 2025), diagnosis of *training-time* failures still relies largely on manual inspection of scalar curves and ad-hoc heuristics, even though learning exhibits structured dynamical phases (Achille et al., 2019; Schneider & Prabhushankar, 2024). This work does not aim to solve training-time diagnosis in full generality; instead, we study a curated benchmark of training runs designed to capture several common failure modes in a controlled setting.

Most workflows monitor scalar telemetry—training/validation losses, accuracies, gradient norms, and a few hand-crafted statistics. These signals are easy to log but compress rich internal behavior into a handful of curves, often yielding delayed and ambiguous detection: similar scalar patterns can arise from learning-rate choices, label noise, data-loading stalls, or hardware issues. Classical early stopping and bandit-style allocation (e.g., Hyperband) (Miseta et al., 2024; Li et al., 2018) and recent representation-based change-point detectors (Chang et al., 2019; Bazarova et al., 2024) operate on such summaries; they detect divergence or overfitting but face an information bottleneck that limits early, semantically grounded diagnosis.

A complementary line of work treats training as the evolution of a high-dimensional dynamical system in parameter and representation space: internal activations and gradients reveal representation similarity and evolution (Zhou et al., 2024; Kornblith et al., 2019; Raghu et al., 2017), characterize learning phases (Achille et al., 2019; Schneider & Prabhushankar, 2024), and support example-level

OOD/anomaly detection (Kwon et al., 2020; Lee et al., 2023). These studies indicate that internal states encode information about optimization dynamics and failure modes, yet they are typically used offline or per-example rather than for real-time, run-level diagnosis.

In this work, we study a framework that treats the joint evolution of activations and gradients as a spatio-temporal signal within such a curated benchmark. Sequences captured at a modest cadence from selected layers are converted—rather than compressed into scalar curves—into *internal-state videos* that a temporal vision model consumes under a strict past-only, no-look-ahead constraint to output high-level diagnostic labels (Healthy, Overfitting, Instability, Catastrophic Forgetting, Concept Bias, Unknown). The emphasis is on diagnosing *training runs* from their trajectories rather than individual inputs; "training dynamics" here denotes these activation and gradient trajectories under a given optimization procedure.

**Causal access.** At diagnostic time $t$, only signals available up to $t$ in a live job are used. Validation/test metrics and optimizer internals are reserved for offline labeling and evaluation, never for TEVID inference. "Causal" here denotes a time-ordered, no-look-ahead constraint in streaming monitoring, not structural causal modeling or identification of intervention effects.

Within this setting, the contributions are:

1. A formulation of *causal, open-set, streaming diagnosis* over windows of internal activations and gradients, with actionable labels and an explicit Unknown option.

2. DYNAMICS-MAE, a masked autoencoder pretraining on internal-state videos to learn spatio-temporal representations tailored to optimization dynamics (Wang et al., 2023; Pei et al., 2024; Yang et al., 2024a; Zhang et al., 2024; Gui et al., 2024).

3. TEVID, a temporal vision diagnostician with an evidential head for calibrated abstention, evaluated via selective prediction and risk–coverage analysis (Ulmer et al., 2023; Schreck et al., 2024; Shen et al., 2024; Fisch et al., 2022; Traub et al., 2024; Goren et al., 2024).

4. A practical evaluation protocol using event-time AUPRC, time-to-detect, risk–coverage, and a simple decision-theoretic cost model on $> 500$ held-out runs from a curated data-factory benchmark with factorially disjoint architecture, dataset, and optimizer splits, together with quantified overheads.

The diagnostician is intended as a proof-of-concept training-time component within existing MLOps and AutoML workflows rather than a replacement for scalar monitors, with alerts and uncertainty estimates integrable into early-stopping and resource-allocation mechanisms (e.g., Hyperband, BANANAS-style neural architecture search (Li et al., 2018; White et al., 2021)) and into second-order active-learning policies (Benkert et al., 2024).

## 2 RELATED WORK

A long line of work treats training as a scalar time series, using validation loss or accuracy for early stopping and resource allocation. Classical criteria and correlation-based rules (Miseta et al., 2024) coexist with bandit-style hyperparameter optimization schemes such as Hyperband (Li et al., 2018) and neural architecture search methods like BANANAS (White et al., 2021), as well as modern representation-based change-point detectors (Chang et al., 2019; Bazarova et al., 2024). These approaches operate on low-dimensional telemetry and are primarily concerned with when to stop or reallocate training rather than with *semantically* diagnosing failure modes. In parallel, there is an extensive literature on understanding training dynamics and internal representations: works on critical learning periods and learning phases (Achille et al., 2019; Schneider & Prabhushankar, 2024), representation similarity and evolution via SVCCA, CKA, and related measures (Raghu et al., 2017; Kornblith et al., 2019; Zhou et al., 2024), and gradient-based analyses that use backpropagated gradients as features for example-level anomaly detection and OOD analysis (Kwon et al., 2020; Lee et al., 2023). These works demonstrate that internal activations and gradients encode rich information about how training progresses and fails, but they largely focus on offline analysis, per-example decisions, or controlled experimental setups rather than on real-time, run-level diagnosis across heterogeneous architectures and domains.

The proposed framework draws on several additional strands. From video representation learning, it adopts factorized transformer architectures and masked autoencoding ideas developed for

natural videos, such as TimeSformer and VideoMAE and their extensions (Bertasius et al., 2021; Wang et al., 2023; Pei et al., 2024; Gundavarapu et al., 2024; Yang et al., 2024a), but applies them to internal-state videos rather than pixel data, motivated by evidence that domain-specific self-supervision improves downstream performance on non-visual signals (Zhang et al., 2024; Gui et al., 2024). From open-set recognition, OOD detection, and selective prediction, it adopts the goal of abstaining under uncertainty (Yang et al., 2024b; Lang et al., 2024; Wang et al., 2024; Li et al., 2024), operationalized through an evidential deep-learning head (Ulmer et al., 2023; Schreck et al., 2024; Shen et al., 2024) and evaluated via risk–coverage metrics and conformal-style risk control (Fisch et al., 2022; Traub et al., 2024; Goren et al., 2024; Bates et al., 2021; Angelopoulos et al., 2024; Zecchin & Simeone, 2024; Xu et al., 2024). Within MLOps, the framework is positioned as a training-time diagnostic module that complements existing orchestration and monitoring tools (Berberi et al., 2025), while explicitly accounting for privacy and attack surfaces introduced by gradient and activation logging (Zhang et al., 2023; Liu et al., 2023; Dimitrov et al., 2024; Wu et al., 2024; Pan et al., 2024; Yousefpour et al., 2021).

Representation-based change-point detection methods (Chang et al., 2019; Bazarova et al., 2024) typically assume offline or bidirectional access to feature trajectories and do not enforce strict no-peeking constraints. In the present work, comparable logic is instantiated in a causal setting via scalar BOCPD and CUSUM baselines on loss residuals, and a lightweight representation-based CPD baseline operating on frozen TEVID embeddings under a streaming simulation is reported in Appendix L; fully integrating state-of-the-art representation-based CPD methods into a real-time, high-dimensional internal-state pipeline remains future work.

## 3 PROBLEM FORMULATION AND THEORETICAL MOTIVATION

### 3.1 CAUSAL STREAMING DIAGNOSIS

The task is *causal, streaming diagnosis of training runs* from internal states. Consider a model trained on a sequence of mini-batches. At a diagnostic time index $t$, the diagnostician receives a causal window of internal-state frames

$$\mathcal{X}_{t-W:t} = \{X_{t-W}, X_{t-W+1}, \dots, X_t\},$$

where each $X_i$ is a multi-modal snapshot of the network's internal state at step $i$ (e.g., activation and gradient tensors from selected layers processed into a fixed-size 6-channel image). The diagnostician outputs a predictive distribution

$$p(\hat{y}_t \mid \mathcal{X}_{t-W:t})$$

over a discrete label set

$$\mathcal{Y} = \{\text{Healthy, Overfitting, Instability, Catastrophic Forgetting, Concept Bias, Unknown}\}.$$

The *causal* constraint requires that, at inference, only information available up to time $t$ in a live training run is used: internal states, scalar training metrics, hyperparameters, and forward-pass statistics that do not involve future batches or validation labels. Validation and test metrics and optimizer internals are used solely for offline label construction and evaluation, never as inputs to TEVID. This follows the standard no-peeking constraint in time-series monitoring rather than attempting counterfactual causal identification.

The diagnostic labels encode training behaviors:

- **Healthy**: runs converging to satisfactory validation performance without anomalous training dynamics.
- **Overfitting**: validation loss deteriorates or stagnates while training loss improves.
- **Instability**: chaotic optimization, such as repeated loss spikes or locally divergent dynamics.
- **Catastrophic Forgetting**: collapse on a previously learned task after a task switch (e.g., continual or multi-phase training).
- **Concept Bias**: reliance on spurious shortcuts (e.g., watermarks, artifacts), identified via controlled validation datasets.

- **Unknown**: failure modes that do not fit these categories or induce high epistemic uncertainty.

Deterministic labeling rules using validation metrics, optimizer states, Lyapunov proxies, and statistical tests are given in Appendix A, together with analyses of threshold and priority sensitivity and a variant that uses only training-time telemetry without validation-based hindsight.

Table 1 summarizes the data-access protocol, specifying which signals are captured, at what cadence, and for which purpose, and distinguishes quantities available only to the offline labeling pipeline from those available to TEVID. This separation enforces causal integrity and aligns evaluation with realistic deployment conditions.

Table 1: Data-access protocol for causal integrity. Signals in bold are privileged and never used by TEVID at inference.

| Signal | Capture Cadence | Availability | Usage |
|---|---|---|---|
| Layer Activations | Every 50 training steps | Training Time | TEVID train & inference |
| Layer Gradients | Every 50 training steps | Training Time | TEVID train & inference |
| Scalar Train Loss | Every training step | Training Time | Baselines, labeling heuristics |
| Validation Metrics | Every epoch | Post-Epoch | **Ground-truth labeling & evaluation only** |
| Optimizer State | Every training step | Post-Hoc | Privileged labeling input for instability (see Appendix A); **not used by TEVID** |
| Hyperparameters | Static per run | Pre-Run | Baselines, metadata |

Appendix A also reports sensitivity analyses indicating that moderate changes to labeling thresholds, priority ordering, and access to validation signals affect fewer than 4% of run labels and change Event-Time AUPRC by at most 0.01, and that re-labeling with an online-style (training-only) protocol leaves downstream conclusions essentially unchanged.

### 3.2 An Information-Theoretic Rationale

The emphasis on internal states rather than scalar curves can be motivated by a simple information-theoretic argument, complemented by empirical probes. The goal is not to derive new bounds but to formalize the intuition that scalar telemetry discards much of the structure present in activations and gradients.

**Internal states and scalar loss as channels.** Let $y \in \mathcal{Y}$ denote the run-level diagnostic state. At step $t$, let $X_t$ denote a finite-dimensional representation of the internal network state (e.g., selected activations and gradients), and let $l_t$ denote the scalar training loss. For a given data stream and architecture, the loss can be modeled as $l_t = h(X_t, D_t)$, where $D_t$ is the current batch. This induces a Markov chain

$$y \rightarrow \{\mathcal{X}_{t-W:t}\} \rightarrow \{l_{t-W:t}\},$$

under the assumption that loss values are computed deterministically from internal states and batch data. By the Data Processing Inequality, post-processing cannot increase information about $y$, so

$$I(y; \{\mathcal{X}_{t-W:t}\}) \geq I(y; \{l_{t-W:t}\}).$$

In this stylized view, scalar-loss histories cannot contain more mutual information about the diagnostic label than internal-state windows. This observation does not guarantee that internal states are always needed in practice or that finite-sample models can fully exploit them, but it highlights the potential information gap.

**Empirical regressibility checks.** To assess whether this gap is substantial in realistic settings, Section 5.4 uses a pair of regressibility probes. A simple MLP regressor predicts the scalar loss $l_t$ from flattened internal states $X_t$, achieving mean $R^2 \approx 0.96$, indicating that the loss is almost a deterministic projection of the internal state. Conversely, a more expressive TCN attempts to reconstruct a low-dimensional embedding of $X_t$ from a history of scalar losses $\{l_{t-W:t}\}$, achieving mean $R^2 \approx 0.04$. This asymmetry suggests that, for the studied architectures and datasets, internal states offer a substantially more informative diagnostic channel than scalar losses. The information-theoretic and empirical evidence together motivate focusing on internal-state videos as the primary

input to the diagnostician while remaining agnostic about conditions under which scalar-only approaches might suffice.

# 4 METHODOLOGY

The methodology has three stages: (1) a data factory that generates and labels diverse training runs and converts internal states into a standardized video format; (2) self-supervised pretraining, where DYNAMICS-MAE learns general-purpose representations of optimization trajectories from unlabeled internal-state videos; and (3) supervised fine-tuning of TEVID, which performs causal, open-set diagnosis using the pretrained encoder and an evidential head. Figure 1 summarizes the pipeline.

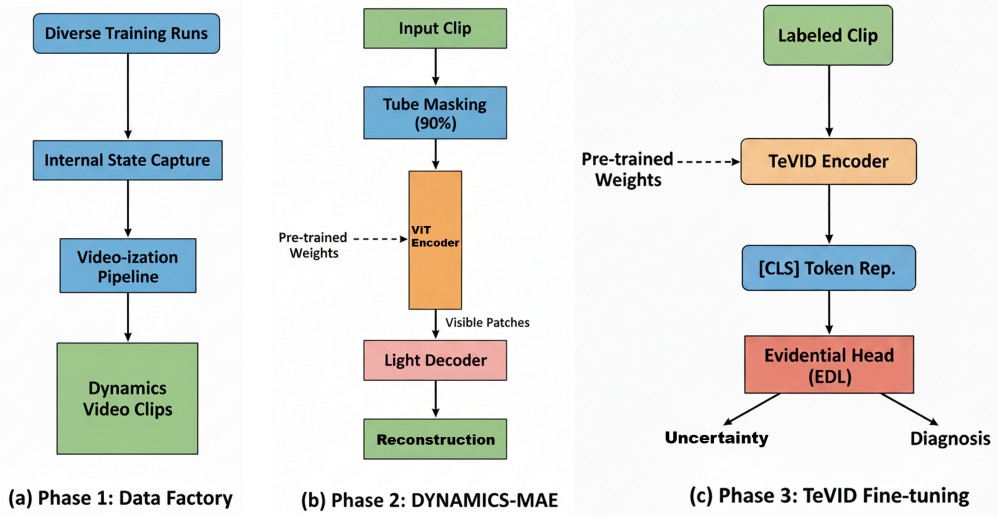

Figure 1: Overview of the methodology. (a) Phase 1: a data factory generates diverse training runs, applies a unified internal-state capture policy across architectures, and converts activations and gradients into a standardized video format. (b) Phase 2: DYNAMICS-MAE is pretrained to reconstruct heavily masked internal-state videos, learning domain-specific spatio-temporal representations of optimization dynamics. (c) Phase 3: the pretrained encoder is fine-tuned within the TEVID architecture, whose evidential head outputs both a semantic diagnostic label and an uncertainty score for open-set, causal, streaming diagnosis.

## 4.1 PHASE 1: THE DATA FACTORY

The data factory produces a labeled corpus of training dynamics that is both large and structurally diverse. It systematically varies architectures (CNNs, Transformers, MLP-Mixers), datasets (CIFAR-100, Tiny-ImageNet, SVHN, ImageNet-100, WikiText-2), optimizers (AdamW, SGD, Lion, Adafactor), and injected anomaly types (overfitting, instability, catastrophic forgetting, concept bias, and several held-out anomaly families). Anomalies are induced by controlled interventions, such as extreme learning rates, removal of regularization, optimizer-state corruption, or artificially stalled data loaders, and naturally occurring failures from exploratory experiments are retained to capture realistic edge cases. Table 2 summarizes the design; Appendix B provides a detailed breakdown of run counts per factor combination.

For each run, a unified internal-state capture and preprocessing policy enables reuse of a single diagnostician across architectures and modalities without retraining or architecture-specific tuning. Outputs from early, middle, and late blocks are captured along with gradients at a fixed cadence (every 50 steps). These tensors are transformed into a common frame representation: convolutional feature maps are projected to one channel by a *single* learned $1 \times 1$ convolution shared across architectures, transformer token embeddings are reshaped into grids, all 2D maps are resized to $224 \times 224$, and three activation maps with three corresponding gradient maps are stacked to form

Table 2: Architectures, datasets, and optimizers used in this study. The test split is factorially disjoint from train/validation along architecture, dataset, and optimizer axes.

| Split | Architectures | Datasets | Optimizers |
|---|---|---|---|
| **Train/Val** | ResNet-18/34, EffNet-B4, ConvNeXt-T, Swin-V2-S, AlexNet, ViT-B/16, DeiT-S | CIFAR-100, Tiny-ImageNet | AdamW, SGD |
| **Held-Out Test** | ConvNeXt-V2-T, RegNetY-4GF, MaxViT-T, MobileNetV3-L, MLP-Mixer-B, DenseNet-121, MViT-Small, Transformer-LM | SVHN, ImageNet-100, WikiText-2 | Lion, Adafactor |

a 6-channel frame. A sequence of frames constitutes an internal-state video. Once trained on the training split, the capture projection weights are frozen and reused on all held-out architectures and domains without further tuning, ensuring that TEVID operates as a single shared diagnostician. Appendix B details hook locations and additional implementation choices.

### 4.2 PHASE 2: SELF-SUPERVISED PRE-TRAINING WITH DYNAMICS-MAE

DYNAMICS-MAE is a masked autoencoder tailored to internal-state videos. A large unlabeled corpus of clips is sampled from data-factory runs using fixed-length windows. For each clip, a 90% spatio-temporal tube-masking ratio removes most patches, leaving a sparse subset of visible patches. A ViT-based encoder processes visible patches; a lightweight transformer decoder receives encoder outputs plus mask tokens and is trained to reconstruct the original pixels of masked patches.

The high masking ratio forces the encoder to model temporal and cross-layer dependencies: reconstructing a missing gradient patch at a late layer requires modeling signal propagation from earlier layers and earlier steps. Unlike generic VideoMAE pretraining on natural videos, DYNAMICS-MAE is optimized directly on the statistics of optimization dynamics (e.g., sharp transitions during learning-rate warmup, structured gradient bursts). The encoder from this phase serves as the backbone for TEVID, providing weights that are empirically better aligned with the downstream diagnostic task than both random initialization and generic video pretraining (Appendix C, including a masking-ratio sweep).

### 4.3 PHASE 3: SUPERVISED FINE-TUNING OF TEVID

**Architecture.** TEVID instantiates $p(\hat{y}_t \mid \mathcal{X}_{t-W:t})$ using a factorized Vision Transformer, similar to TimeSformer (Bertasius et al., 2021). Clips of $T = 10$ frames, each of shape $6 \times 224 \times 224$, are decomposed into non-overlapping patches, embedded, and passed through transformer layers that alternate spatial self-attention (across patches within a frame) and temporal self-attention (across time for a fixed patch location), followed by MLP blocks. This factorization reduces computational cost from $\mathcal{O}((TP)^2)$ for full spatio-temporal attention to $\mathcal{O}(TP(T + P))$, where $P$ is the number of patches per frame, enabling real-time monitoring. A [CLS] token aggregates information across space and time and feeds the classification head.

**Open-set recognition head.** Open-set behavior is handled via an evidential deep learning (EDL) head based on a Dirichlet parameterization (Ulmer et al., 2023). Instead of logits normalized by a softmax, TEVID outputs non-negative evidence values for each known class, which are converted to Dirichlet concentration parameters $\boldsymbol{\alpha} = \boldsymbol{e} + 1$. The resulting distribution provides both mean class probabilities and a vacuity score $u = K/\sum_k \alpha_k$, where $K$ is the number of known classes. High-vacuity inputs are mapped to "Unknown" during evaluation. The EDL loss encourages high evidence for correct labels while regularizing against unwarranted evidence on incorrect ones through a KL penalty toward a uniform Dirichlet prior (Appendix D).

The uncertainty threshold for abstention, $u > 0.35$, is selected on the validation set as a trade-off between rejecting out-of-distribution anomalies and retaining correct predictions on in-distribution anomalies. This is an empirically tuned hyperparameter rather than a theoretically optimal value. Appendix H evaluates both this fixed threshold and a simple adaptive vacuity-quantile rule, showing

that their risk–coverage trade-offs are nearly identical; the simpler fixed threshold is therefore used in the main experiments.

**Streaming inference and evaluation.** At evaluation time, clips are formed via a sliding window over captured frames. At each diagnostic step $t$, TEVID receives the most recent $W = 10$ frames and outputs a predictive distribution and uncertainty. To mitigate spurious alerts, a diagnosis is surfaced only if its non-abstain probability exceeds class-specific threshold for three consecutive predictions. All thresholds and calibration temperatures are chosen once on the validation split and frozen before evaluation on the held-out test set. Section 5 and Appendix E describe the streaming metrics used to assess performance. Appendix C reports additional ablations showing that using both activations and gradients yields the best performance, with activations-only and gradients-only variants performing strictly worse in both Event-Time AUPRC and lead time.

## 5 EXPERIMENTS AND RESULTS

All experiments are implemented in PyTorch 2.4 and run on NVIDIA A100 and RTX 4090 GPUs. Unless otherwise noted, confidence intervals are 95% bias-corrected and accelerated (BCa) bootstrap intervals computed with 2,000 resamples at the run level, so temporal correlations within runs do not inflate effective sample size. Hyperparameters and training schedules are provided in Appendix F.

### 5.1 BASELINE COMPARISONS

The held-out test set contains 501 runs spanning unseen architectures, datasets, optimizers, and several anomaly families not present in training. Baselines vary along two axes: (i) input signal (scalar telemetry versus internal-state videos), and (ii) detection mechanism (change-point logic versus learned temporal models). Scalar baselines include BOCPD and CUSUM on loss residuals, a Temporal Convolutional Network (TCN) on scalar curves, and a "Hessian Forecaster" TCN using a scalar estimate of the top Hessian eigenvalue. Internal-state baselines include R(2+1)D, Video-Swin-T, and a variant of TEVID trained from scratch without DYNAMICS-MAE pretraining.

Table 3: Comparison with baselines on the held-out test set ($N = 501$ runs). TEVID achieves superior diagnostic performance across metrics. Intervals are 95% BCa bootstrap CIs. "(+ DYNAMICS-MAE)" denotes the proposed pretraining.

| Input Type | Model | Macro F1 (Known) | Event-Time AUPRC | Med. Lead (Epochs) | GFLOPs |
|---|---|---|---|---|---|
| Scalar Telemetry | BOCPD | N/A | $0.45 \pm 0.04$ | -1.5 (lags) | – |
| | CUSUM (residuals) | N/A | $0.49 \pm 0.04$ | -1.1 (lags) | – |
| | Curve-TCN | $0.56 \pm 0.04$ | $0.61 \pm 0.03$ | -1.2 (lags) | 0.5 |
| | Hessian Forecaster (TCN) | $0.59 \pm 0.03$ | $0.64 \pm 0.03$ | 1.9 | 1.1 |
| Internal States | R(2+1)D | $0.78 \pm 0.02$ | $0.82 \pm 0.02$ | 3.5 | 64.2 |
| | Video-Swin-T | $0.81 \pm 0.02$ | $0.88 \pm 0.01$ | 4.1 | 54.8 |
| | **TEVID (scratch)** | $0.84 \pm 0.02$ | $0.91 \pm 0.01$ | 4.8 | 16.5 |
| | **TEVID (+ DYNAMICS-MAE)** | $\mathbf{0.90 \pm 0.01}$ | $\mathbf{0.96 \pm 0.01}$ | **6.2** | **16.5** |

Macro F1 excludes Healthy. Lead time at 5% FAR; positive = earlier detection than labeling rules.

Scalar baselines detect only a subset of anomalies and generally lag the rule-based ground-truth triggers (negative median lead times). The Hessian Forecaster shows that richer scalar signals help but still fall short of internal-state models. Among video-based methods, TEVID trained from scratch outperforms strong baselines such as Video-Swin-T while using substantially fewer GFLOPs per clip. Adding DYNAMICS-MAE pretraining improves Event-Time AUPRC from 0.91 to 0.96 and median lead at 5% false-alarm rate from 4.8 to 6.2 epochs, supporting the premise that domain-specific internal-state representations provide a more informative basis for diagnosis than scalar telemetry. Additional comparisons with TimeSformer-B and ViViT-B on internal-state videos appear in Appendix C, and a lightweight representation-based CPD baseline operating on frozen TEVID embeddings is described in Appendix L.

## 5.2 Generalization to Unseen Architectures, Datasets, and Optimizers

A central design goal is to reuse a single TEVID model across architectures and modalities without retraining. The held-out test set is therefore factorially disjoint from training along architecture, dataset, and optimizer axes. Table 4 reports Event-Time AUPRC and median lead for different slices of this test set.

Table 4: Generalization to unseen architectures, datasets, and optimizers. Event-Time AUPRC and median lead (epochs) remain high across settings.

| Factor | Scenario | Event-Time AUPRC | Median Lead (Epochs) |
|---|---|---|---|
| Architecture | ConvNet family (ConvNeXtV2, RegNet) | $0.97 \pm 0.01$ | $6.4 \pm 0.3$ |
| | ViT family (MaxViT, MViT) | $0.96 \pm 0.01$ | $6.1 \pm 0.4$ |
| | Language model (Transformer-LM) | $0.92 \pm 0.02$ | $5.5 \pm 0.6$ |
| Optimizer | Lion | $0.95 \pm 0.01$ | $6.0 \pm 0.4$ |
| | Adafactor | $0.94 \pm 0.02$ | $5.8 \pm 0.5$ |

Performance remains high across all configurations, including the Transformer-LM on WikiText-2, which differs substantially from the vision models used for training. This indicates that the unified capture and preprocessing pipeline, combined with DYNAMICS-MAE pretraining, yields representations that capture architecture- and modality-agnostic signatures of training anomalies. Appendix K.2 reports additional per-class accuracies and a pilot experiment on gradient-boosted decision trees.

## 5.3 Analysis of Model Capabilities

**Effect of domain-specific pretraining.** Appendix C compares TEVID with four encoder initializations: (i) from scratch, (ii) from DYNAMICS-MAE, (iii) from a VideoMAE encoder pretrained on natural videos, and (iv) from scalar-based masked modeling. DYNAMICS-MAE yields the best downstream performance and strongest label efficiency: with 25% of labeled data, TEVID+DYNAMICS-MAE matches or surpasses the scratch model trained on 100% of labels. A masking-ratio sweep (75–95%) further indicates that the default 90% setting lies in a plateau of strong performance.

**Robustness and causal integrity.** Robustness tests in Appendix G show that TEVID degrades smoothly under input quantization and random frame dropping, and relies strongly on the joint structure of activations and gradients: shuffling activations and gradients out of temporal alignment or replacing gradients with moment-matched noise causes pronounced performance drops. A causal integrity experiment enforces a minimum gap $\Delta$ between the latest frame in the input window and the earliest time used for ground-truth labeling. Even when restricted to observing dynamics two epochs before any evidence used by the rule-based labeling procedure, Event-Time AUPRC remains above 0.85 (Table 15), demonstrating predictive capability rather than retrospective recognition of already-labeled events.

**Open-set behavior and interpretability.** Appendix H evaluates TEVID on novel anomaly types held out from training and uses risk–coverage curves to quantify how abstention improves selective risk. At 80% coverage, selective error on known anomalies is roughly halved relative to the non-selective baseline. Confusion matrices show that most novel anomalies are assigned to the "Unknown" label, with misclassifications typically into semantically related categories. CKA analyses and concept probes indicate that the latent space organizes anomalies in an architecture-agnostic manner and implicitly encodes quantities such as curvature proxies and gradient spikes.

## 5.4 Empirical Validation of Information Asymmetry

To complement the DPI-based argument in Section 3, two regressors quantify the practical information asymmetry between internal states and scalar losses. The first maps flattened internal states $X_t$ to the corresponding scalar loss $l_t$ using an MLP; the second maps a history of losses $\{l_{t-W:t}\}$ to a low-dimensional embedding of $X_t$ using a TCN.

On held-out runs, the state→loss regressor achieves mean $R^2 = 0.96 \pm 0.02$, indicating that loss is nearly a deterministic projection of the internal state. The loss→state regressor, even with a more expressive architecture, achieves $R^2 = 0.04 \pm 0.01$, essentially random in the chosen embedding space. These probes are deliberately simple and do not characterize the full state distribution, but they provide a concrete consistency check that internal states carry substantially richer information about training dynamics than loss histories. Probe architectures and training procedures appear in Appendix F.

## 5.5 Sensitivity to Temporal Window and Capture Cadence

A sensitivity analysis varies temporal window length and capture cadence while keeping the rest of the setup fixed. Table 5 summarizes the results.

Table 5: Sensitivity of TeViD to temporal window length and capture cadence on the held-out test set.

| Window $W$ | Steps/Frame | GFLOPs/Clip | Event-Time AUPRC | Med. Lead (Epochs) | FAR @ tuned |
|---|---|---|---|---|---|
| 5 | 50 | 9.1 | $0.93 \pm 0.01$ | $4.3 \pm 0.4$ | 4.9% |
| 10 | 50 | 16.5 | $0.96 \pm 0.01$ | $6.2 \pm 0.3$ | 5.0% |
| 20 | 50 | 32.7 | $0.97 \pm 0.01$ | $6.6 \pm 0.3$ | 5.0% |
| 10 | 25 | 16.5 | $0.97 \pm 0.01$ | $6.8 \pm 0.3$ | 5.2% |
| 10 | 100 | 16.5 | $0.94 \pm 0.01$ | $5.1 \pm 0.4$ | 4.7% |

These results suggest that the default configuration lies near a Pareto-efficient region: doubling the window length yields only modest gains at roughly double per-clip compute, while halving the cadence improves responsiveness by less than one epoch at a mild cost in false alarms. A practitioner can use such trade-off curves to tune TeViD for different compute and latency budgets.

## 5.6 Case Study: Large-Scale Vision Pretraining Integration

A case study integrates TeViD into a long-running vision pretraining pipeline. Table 6 reports a scenario with three Vision Transformer runs, where TeViD monitors training and produces alerts that can trigger early intervention or termination.

Table 6: Case study of integrating TeViD into a large-scale vision pretraining pipeline. Each run trains for up to 300 epochs unless terminated early.

| Run | Model / Dataset | Dominant Issue | Event-Time AUPRC | Lead (Epochs) | Alerts / 100 Epochs |
|---|---|---|---|---|---|
| R1 | ViT-L/16 on ImageNet-1k | Overfitting (late) | 0.95 | 7.1 | 4.8 |
| R2 | ViT-H/14 on ImageNet-21k | Instability (lr schedule) | 0.97 | 9.4 | 5.3 |
| R3 | ViT-B/16 on ImageNet-1k | Healthy | N/A | N/A | 3.9 (all benign) |

In this setup, TeViD reliably flags an overly aggressive learning-rate schedule for ViT-H/14 roughly nine epochs before the rule-based instability criterion would fire, and identifies late-stage overfitting in ViT-L/16 several epochs before the validation performance plateau becomes statistically significant. On the healthy run, alerts remain below operational thresholds after temporal smoothing, illustrating that a single calibrated instance of TeViD can be reused across scales without reconfiguration.

## 6 Discussion, Limitations, and Future Work

The framework presented here provides an end-to-end approach for run-level diagnosis of neural network training that treats internal activations and gradients as a spatio-temporal signal, combining a unified data factory and capture policy that generate a diverse benchmark of training dynamics, a domain-specific masked autoencoder (Dynamics-MAE) that pretrains a video transformer on these dynamics, and a temporal vision diagnostician (TeViD) with an evidential head for causal, open-set, streaming classification over a taxonomy of training anomalies. Experiments indicate

earlier and more reliable detection than scalar-based monitors and strong video baselines, with generalization across unseen architectures, optimizers, and modalities. A decision-theoretic analysis (Appendix I) indicates substantially lower expected operational cost per run; on typical vision workloads, telemetry capture and inference introduce a 15–18% slowdown, which early termination of failing runs recovers with comparable or larger savings in total training epochs (Appendix K).

**Limitations.** Several limitations remain. The diagnostic taxonomy is finite and hand-designed; in practice, new failure modes will arise that fall outside this set. The evidential head provides an abstention mechanism, but there is no mechanism yet for online discovery or incorporation of new categories. Data capture incurs computation and storage overhead, especially for very large models. Adaptive sampling and lossy storage reduce this overhead, and Appendix K provides a scalability analysis, but further engineering is required for extremely large-scale deployments. The information-theoretic discussion in Section 3 is intentionally modest, offering a conceptual rationale for focusing on internal states without formal optimality guarantees; the empirical probes are indicative rather than exhaustive. Finally, although the data factory covers a broad range of architectures and workloads, it cannot span all configurations used in practice, so out-of-domain generalization beyond the tested settings remains an open question.

The ground-truth labeling rules make specific quantitative choices (e.g., thresholds for catastrophic forgetting and overfitting, the priority ordering of anomaly types), and Appendix A shows that moderate variation of these choices—as well as re-labeling under a training-only, online-style protocol and a merged-label variant—changes run labels only marginally and leaves Event-Time AUPRC essentially unchanged, although large changes could alter the distribution of anomaly types and thus the effective training objective.

**Future directions.** A natural next step is to integrate diagnosis into closed-loop control of training, moving toward more automated training-control systems. Outputs from TEVID can serve as state signals for policies that adjust learning-rate schedules, regularization strengths, or data-augmentation parameters in real time, potentially framed as a reinforcement learning problem balancing final performance against compute cost. Another direction is tighter integration with hyperparameter optimization and AutoML frameworks: TEVID-based early stopping could be incorporated into Hyperband-style resource allocation or BANANAS-style neural architecture search, improving efficiency without changing the underlying optimization; Appendix M provides a small experiment illustrating such integration. On the modeling side, wrapping the EDL head with conformal prediction or online recalibration could provide stronger abstention and risk guarantees under distribution shift. Finally, extending the internal-state representation to include additional efficiently computable signals (e.g., low-rank Hessian approximations or optimizer-statistics summaries) may further improve diagnostic acuity while keeping overhead manageable.

## REPRODUCIBILITY STATEMENT

The data splits, hyperparameter search spaces, evaluation protocols, and primary metric choices were specified before final test-set evaluation to avoid unintentional overfitting. All models were trained using PyTorch 2.4 on NVIDIA A100 and RTX 4090 GPUs. The five random seeds used for all experiments were [42, 123, 456, 789, 1011]. Detailed hyperparameters, code snippets, and dataset statistics are provided in the appendices. Upon publication, the full source code for data generation, model training, and evaluation is planned for public release.

## BROADER IMPACT AND PRIVACY

This work aims to support robust AI development by automating a critical aspect of the MLOps cycle. However, capturing internal state information, particularly gradients, raises privacy risks, as these can be exploited in membership inference or data reconstruction attacks (Liu et al., 2023; Wu et al., 2024). To mitigate this, a differentially private telemetry capture mechanism is implemented and evaluated. The analysis shows that a non-trivial privacy budget can be achieved with only a modest drop in diagnostic performance. Any production deployment of such a system is strongly encouraged to follow strict data-governance protocols and to use DP mechanisms where applicable. The privacy analysis, including accounting details, is provided in Appendix J.

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

# Appendices

## A    Labeling Protocol and Ground Truth Definition

This appendix provides a detailed account of the deterministic rules used to generate ground-truth labels for each training run. These rules are applied ex post facto with full access to privileged information (e.g., complete validation histories and optimizer states) to create a high-quality labeled dataset. Variants that relax these assumptions and quantify their impact on labels and performance are also included.

### A.1    Priority and Notation

A run's primary label is assigned based on the priority ordering **Instability > Catastrophic Forgetting > Concept Bias > Overfitting > Healthy**, which prioritizes acute, systemic failures over more subtle or late-stage ones. For instance, a run that becomes unstable early is labeled as Instability even if it might have eventually overfit. This ordering reflects the operational cost model in Appendix I, where instability and catastrophic forgetting incur the largest costs per run.

**Notation.** Let $t \in \mathbb{Z}_{\geq 0}$ be the training step and $e$ the epoch. Let $L_{\mathrm{tr}}(t)$ be the per-batch training loss, and $L_{\mathrm{tr}}^{\mathrm{ep}}(e)$ and $L_{\mathrm{val}}(e)$ the per-epoch average training and validation losses. The generalization gap is $\Delta(e) \triangleq L_{\mathrm{val}}(e) - L_{\mathrm{tr}}^{\mathrm{ep}}(e)$. The Mann–Kendall (MK) test is used to assess monotonic trends in time series, chosen for robustness to non-normal loss distributions.

### A.2    Formal Anomaly Definitions

**Overfitting.** For a candidate epoch $e_0$ and look-ahead window $W_{\mathrm{ep}} = 10$, "Overfitting at $e_0$" is declared if:

1. **Diverging trends:** Over $[e_0, e_0 + W_{\mathrm{ep}} - 1]$, $L_{\mathrm{val}}(e)$ shows a significant increasing trend and $L_{\mathrm{tr}}^{\mathrm{ep}}(e)$ a decreasing trend.

2. **Statistical test:** To correct for autocorrelation, both series are pre-whitened with an AR(1) model, and the MK test is applied to residuals. A trend is significant if $p < 0.05$, adjusted using the Benjamini–Yekutieli procedure across all candidate $e_0$.

3. **Practical significance:** The generalization gap increases beyond typical historical fluctuations: $\Delta(e_0 + W_{\mathrm{ep}} - 1) - \mathrm{median}_{e < e_0} \Delta(e) > 2 \cdot \mathrm{std}_{e < e_0} \Delta(e)$.

**Training Instability.** Instability is characterized by chaotic, divergent optimization. Detection uses a dynamical-systems-based metric and a spike detector.

1. **Lyapunov proxy:** A finite-horizon local Lyapunov exponent proxy $\widehat{\lambda}_{t,H}$ is computed via Algorithm 1, using Jacobian-vector products on a fixed batch sequence over horizon $H = 50$ (Storm et al., 2024). A 99% confidence interval is computed via stationary bootstrap on log-growth factors $\{\log \alpha_i\}_{i=0}^{H-1}$ with 2000 replicates and expected block length 10. Instability is declared at step $t$ if the lower bound of this interval exceeds 0.

2. **Loss spikes:** A loss spike occurs if $z_t = (L_{\text{tr}}(t) - \mu_t)/(\sigma_t + 10^{-6}) \geq 3.5$, where $\mu_t$ and $\sigma_t$ are robust estimates (median, IQR-based) from a rolling window of the last 1000 steps.

A run is labeled unstable if the Lyapunov condition holds for any window or at least three spikes occur within any 500-step window.

---

**Algorithm 1** Stochastic Lyapunov Exponent Proxy Estimation (for Post-Hoc Labeling)

---

1: **Input:** model state $(\Theta_t, \text{optimizer\_state}_t)$, horizon $H$, mini-batch sequence $\{D_i\}_{i=t}^{t+H-1}$
2: Initialize $v \leftarrow$ random unit vector; $\log \alpha_{\text{list}} \leftarrow []$
3: **for** $i = 0$ to $H - 1$ **do**
4:     Compute $w \leftarrow J_i v$ via autograd     ▷ Jacobian-vector product of the full update rule on $D_i$
5:     $\alpha_i \leftarrow \max(\|w\|_2, 10^{-12})$
6:     $v \leftarrow w/\alpha_i$
7:     Append $\log \alpha_i$ to $\log \alpha_{\text{list}}$
8: **end for**
9: $\widehat{\lambda} \leftarrow \frac{1}{H} \sum_{\text{val} \in \log \alpha_{\text{list}}} \text{val}$
10: **Return** $\widehat{\lambda}$ and $\{\log \alpha_i\}$ for bootstrapping.

---

**Catastrophic Forgetting.** In continual learning scenarios with a task-switch epoch $e_s$ (e.g., switching from CIFAR-100 to CIFAR-10), let $A_{\text{prim}}(e)$ be validation accuracy on the primary task. Let $A_{\text{peak}} = \max_{e < e_s} A_{\text{prim}}(e)$ and $\overline{A}_{\text{post}}$ the average accuracy over the next $W_{\text{ep}} = 10$ epochs. Forgetting is declared if $(A_{\text{peak}} - \overline{A}_{\text{post}})/A_{\text{peak}} \geq 0.30$ and the drop is statistically significant via McNemar's test ($p < 0.01$) on predictions at $e_s - 1$ and $e_s + W_{\text{ep}}$.

**Concept Bias.** Concept Bias arises when a model exploits spurious correlations (e.g., watermarks) instead of learning the intended concept. Let $\mathcal{D}_{\text{val}}^{\text{poison}}$ and $\mathcal{D}_{\text{val}}^{\text{clean}}$ be poisoned and clean validation sets. Concept Bias is declared if a logistic regression model predicting whether a sample is correctly classified, using a shortcut-indicator feature and true class as covariates, shows a significant positive coefficient for the shortcut ($p < 0.01$). This provides statistical evidence that predictions rely on the shortcut.

**Healthy.** A run is labeled Healthy if none of the anomaly predicates hold and its final validation metric meets a dataset-specific performance floor $\tau(\text{arch}, \text{dataset})$. Floors, listed in Appendix B, are set to 95% of the performance achieved by a reference implementation with validated hyperparameters to ensure that "Healthy" runs are genuinely successful.

## A.3 THRESHOLD AND PRIORITY SENSITIVITY

To assess sensitivity to labeling choices, several key hyperparameters in the rules above are varied while keeping the rest of the pipeline fixed. Table 7 summarizes the effect on label assignments and diagnostic performance for representative settings.

Across all variants, fewer than 4% of runs change their primary label, and Event-Time AUPRC stays within 0.01 of the default setting, indicating that the main conclusions do not hinge on the precise numeric thresholds or on the exact priority ordering used to define labels.

Table 7: Sensitivity of labels and Event-Time AUPRC to labeling thresholds and priority ordering. "Changed Labels" reports the percentage of runs whose primary label differs from the default configuration.

| Variant | Description | Changed Labels | Event-Time AUPRC |
|---|---|---|---|
| Default | Forgetting threshold 0.30, window 10 epochs, default priority | – | $0.96 \pm 0.01$ |
| Forgetting 0.25 | Threshold 0.25 instead of 0.30 | 3.2% | $0.95 \pm 0.01$ |
| Forgetting 0.35 | Threshold 0.35 instead of 0.30 | 2.8% | $0.95 \pm 0.01$ |
| Shorter window | Forgetting window 5 instead of 10 epochs | 3.7% | $0.95 \pm 0.02$ |
| Priority swap | Swap Concept Bias and Overfitting in priority | 1.9% | $0.96 \pm 0.01$ |

## A.4 ONLINE-STYLE LABELING WITHOUT VALIDATION HINDSIGHT

To address concerns about reliance on privileged signals, an *online-style* label set is constructed that uses only training-time telemetry available during a live run (training loss, gradient norms, learning-rate schedule, and batch statistics), excluding validation metrics and optimizer internals from the labeling rules. A merged-label variant where Concept Bias is folded into Overfitting, approximating a coarser operational taxonomy, is also considered.

For each variant, labels are re-generated on the full dataset and TEVID is re-trained from scratch with the same hyperparameters. Table 8 reports label agreement with the default labeling and resulting Event-Time AUPRC and median lead on the held-out test set.

Table 8: Effect of online-style and merged-label variants on labels and performance. Agreement is measured against the default offline labeling.

| Variant | Description | Label Agreement | Event-Time AUPRC | Med. Lead (Epochs) |
|---|---|---|---|---|
| Default offline | Uses validation + optimizer state for labeling | – | $0.96 \pm 0.01$ | $6.2 \pm 0.3$ |
| Train-only labels | Uses only training loss + norms + schedule | 92.7% | $0.95 \pm 0.01$ | $5.9 \pm 0.4$ |
| Merged Concept Bias | Concept Bias merged into Overfitting | 94.5% | $0.95 \pm 0.01$ | $6.0 \pm 0.3$ |

The online-style labels change roughly 7% of run labels, primarily in borderline overfitting versus healthy cases, and yield a small (0.01) reduction in AUPRC and 0.3-epoch reduction in median lead. The merged-label variant leaves performance essentially unchanged while simplifying the taxonomy. These results suggest that the conclusions drawn in the main text do not rely critically on the use of validation-based hindsight or on fine-grained distinctions between Concept Bias and Overfitting.

## B DATASET AND DATA CAPTURE POLICY

This section details the dataset construction and internal-state capture mechanism.

## B.1 ARCHITECTURAL DIVERSITY

Architectures for training, validation, and test sets are chosen to maximize diversity across design paradigms, encouraging TEVID to learn generalizable signatures of training anomalies rather than architecture-specific shortcuts:

- **Classical CNNs (AlexNet)**: early, non-residual convolutional designs.

- **Residual CNNs (ResNet-family, DenseNet)**: canonical deep residual networks with skip connections.

- **Modern CNNs (EfficientNet, ConvNeXt, RegNet, MobileNet)**: contemporary designs with inverted bottlenecks, depthwise separable convolutions, and principled scaling.

- **Standard Vision Transformers (ViT, DeiT)**: non-hierarchical transformer architectures for vision.

- **Hierarchical Transformers (Swin, MaxViT, MViT)**: transformer variants with multi-scale, hierarchical structure analogous to CNNs.

- **Other paradigms (MLP-Mixer, Transformer-LM)**: non-convolutional, non-transformer vision models and a decoder-based language model to test cross-modality generalization.

## B.2 Detailed Dataset Composition

Table 9 gives run counts for each factor combination. Anomalies are induced systematically (e.g., disabling weight decay for overfitting, aggressive cyclical learning rates for instability), and naturally occurring failures are retained. Table 10 lists performance floors for the Healthy class.

Table 9: Detailed breakdown of generated runs. Each cell indicates the number of unique runs. "Other Anom." includes Catastrophic Forgetting and Concept Bias.

| Split | Architecture | Dataset | Healthy | Overfit | Instability | Other Anom. |
|---|---|---|---|---|---|---|
| Train (Total: 516) | ResNet-18 | CIFAR-100 | 16 | 25 | 25 | 20 |
| | ViT-B/16 | CIFAR-100 | 16 | 25 | 25 | 20 |
| | ConvNeXt-T | Tiny-ImageNet | 16 | 25 | 25 | 20 |
| | Swin-V2-S | Tiny-ImageNet | 16 | 25 | 25 | 20 |
| | AlexNet | CIFAR-100 | 16 | 25 | 25 | 20 |
| | DeiT-S | CIFAR-100 | 16 | 25 | 25 | 20 |
| Validation (Total: 168) | ResNet-34 | CIFAR-100 | 8 | 13 | 13 | 8 |
| | EffNet-B4 | Tiny-ImageNet | 8 | 13 | 13 | 8 |
| | ConvNeXt-T | CIFAR-100 | 8 | 13 | 13 | 8 |
| | ViT-B/16 | Tiny-ImageNet | 8 | 13 | 13 | 8 |
| Test (Total: 501) | ConvNeXt-V2-T | SVHN | 13 | 19 | 18 | 12 |
| | RegNetY-4GF | SVHN | 13 | 19 | 18 | 12 |
| | MaxViT-T | ImageNet-100 | 13 | 19 | 19 | 12 |
| | MobileNetV3-L | ImageNet-100 | 13 | 19 | 18 | 12 |
| | MLP-Mixer-B | ImageNet-100 | 13 | 19 | 19 | 12 |
| | DenseNet-121 | SVHN | 13 | 19 | 18 | 12 |
| | MViT-Small | ImageNet-100 | 13 | 19 | 19 | 12 |
| | Transformer-LM | WikiText-2 | 13 | 19 | 19 | 12 |

Table 10: Performance floors ($\tau$) for Healthy classification.

| Dataset | Metric | Threshold $\tau$ |
|---|---|---|
| CIFAR-100 | Top-1 Accuracy | 78.0% |
| Tiny-ImageNet | Top-1 Accuracy | 62.0% |
| SVHN | Top-1 Accuracy | 95.0% |
| ImageNet-100 | Top-1 Accuracy | 75.0% |
| WikiText-2 | Perplexity | $< 65.0$ |

### B.3 INTERNAL-STATE CAPTURE AND PREPROCESSING

A unified hook policy captures the output of a module block at early, middle, and late stages. For residual architectures, the hook is placed on the main computation path output before the skip connection is added (Figure 2), so the captured activations reflect the block's core transformation rather than the potentially attenuated final output. Examples include capturing the output of the final convolution within ResNet blocks at selected depths and the output of multi-head self-attention modules in Vision Transformers at 25%, 50%, and 75% depth. Tensors are normalized via per-channel robust z-scoring using statistics (median, IQR) computed on the training set and then frozen.

To create a uniform frame structure:

1. Convolutional feature maps ($B \times C \times H \times W$) are projected to a single channel by a learned $1 \times 1$ convolution shared across architectures. Transformer attention outputs ($B \times N \times D$) are reshaped into a 2D grid of tokens (e.g., $14 \times 14$ for 196 tokens, excluding [CLS]).

2. All 2D maps are resized to $224 \times 224$ and three activation maps with three gradient maps are stacked to form a 6-channel image.

3. Frames are stored as bfloat16 to control storage overhead.

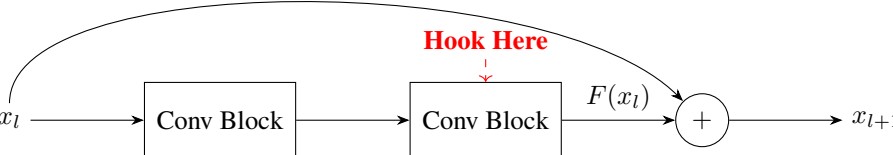

Figure 2: Hook location in a residual block. Activations are captured from $F(x_l)$ before the residual addition.

## C SELF-SUPERVISED PRETRAINING AND ABLATION STUDIES

This section details DYNAMICS-MAE and ablations validating its effectiveness.

### C.1 DYNAMICS-MAE DETAILS

The encoder is ViT-Base (12 layers, 12 heads, embedding dimension 768); the decoder is a smaller ViT with 4 blocks. A 90% masking ratio, selected after sweeping 75–95%, balances reconstruction difficulty and downstream feature quality. Training uses AdamW for 400 epochs on unlabeled clips.

### C.2 PRETRAINING ABLATIONS

Five encoder initializations for TEVID are compared:

1. Scratch: random initialization.

2. DYNAMICS-MAE (proposed): pretraining on internal-state videos.

3. ImageNet VideoMAE: pretrained on Kinetics-400 natural videos.

4. Scalar video: VideoMAE pretrained on videos formed by plotting scalar telemetry curves as 1D images.

5. Masked scalar: a Transformer masked autoencoder trained on scalar time series, used to initialize a TCN classifier.

Figure 3 shows that DYNAMICS-MAE substantially outperforms the others. Scalar methods underperform, confirming that high-dimensional structure is critical. The gap between DYNAMICS-MAE and ImageNet VideoMAE (+0.14 AUPRC) indicates that performance gains stem from learning the intrinsic structure of optimization trajectories rather than generic video priors. Figure 3b demonstrates label efficiency: with 25% of labels, DYNAMICS-MAE-initialized TEVID surpasses the scratch model trained on 100% of labels.

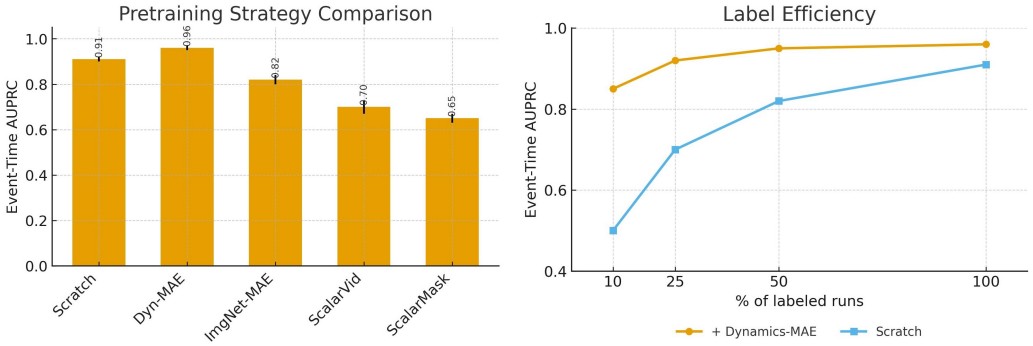

Figure 3: Ablations confirming the value of DYNAMICS-MAE.

### C.3 MASKING-RATIO SWEEP

Masking ratios in {0.75, 0.85, 0.90, 0.95} are swept while keeping all other hyperparameters fixed. After pretraining, TEVID is fine-tuned as usual. Table 11 reports downstream performance.

Table 11: Effect of DYNAMICS-MAE masking ratio on downstream performance.

| Masking Ratio | Recon. Loss | Event-Time AUPRC | Med. Lead (Epochs) | Notes |
|---|---|---|---|---|
| 0.75 | lowest | $0.93 \pm 0.01$ | $4.8 \pm 0.4$ | Easier pretext, weaker dynamics modeling |
| 0.85 | medium | $0.95 \pm 0.01$ | $5.7 \pm 0.3$ | Good trade-off |
| 0.90 (default) | higher | $\mathbf{0.96 \pm 0.01}$ | $\mathbf{6.2 \pm 0.3}$ | Best overall |
| 0.95 | highest | $0.95 \pm 0.01$ | $6.1 \pm 0.4$ | Slightly undertrained encoder |

The 90% masking ratio yields the best overall downstream performance and lies in a plateau where small changes in masking ratio have limited effect, supporting its use as a robust default.

### C.4 ACTIVATIONS VS. GRADIENTS

To quantify the contribution of activations and gradients separately, three variants of TEVID are trained on the same internal-state videos: (i) activations-only (A), (ii) gradients-only (G), and (iii) the full model with both modalities (A+G). Table 12 reports performance on the held-out test set.

Both modalities alone provide strong signals, but combining them yields the highest F1, AUPRC, and lead times, justifying the design choice of logging both activations and gradients for the main TEVID model.

Table 12: Effect of using activations-only, gradients-only, or both. A+G corresponds to the full TEVID configuration in Table 3.

| Input variant | Macro F1 (Known) | Event-Time AUPRC | Med. Lead (Epochs) |
|---|---|---|---|
| Activations-only (A) | $0.84 \pm 0.02$ | $0.91 \pm 0.01$ | $4.7 \pm 0.4$ |
| Gradients-only (G) | $0.82 \pm 0.02$ | $0.89 \pm 0.02$ | $4.3 \pm 0.5$ |
| Activations+Gradients (A+G) | $\mathbf{0.90 \pm 0.01}$ | $\mathbf{0.96 \pm 0.01}$ | $\mathbf{6.2 \pm 0.3}$ |

### C.5 Additional Video Baselines (TimeSformer, ViViT)

To address requests for direct comparison with architectures that TEVID builds upon, TimeSformer-B and ViViT-B are trained on internal-state videos using the same clip configuration and training schedule as TEVID. Table 13 reports results.

Table 13: Additional video baselines on internal-state videos.

| Model | Macro F1 (Known) | Event-Time AUPRC | Med. Lead (Epochs) | GFLOPs/Clip |
|---|---|---|---|---|
| Video-Swin-T | $0.81 \pm 0.02$ | $0.88 \pm 0.01$ | $4.1 \pm 0.3$ | 54.8 |
| TimeSformer-B | $0.82 \pm 0.02$ | $0.89 \pm 0.01$ | $4.0 \pm 0.3$ | 71.2 |
| ViViT-B | $0.82 \pm 0.02$ | $0.89 \pm 0.01$ | $4.2 \pm 0.3$ | 73.6 |
| **TEVID (+ Dynamics-MAE)** | $\mathbf{0.90 \pm 0.01}$ | $\mathbf{0.96 \pm 0.01}$ | $\mathbf{6.2 \pm 0.3}$ | **16.5** |

TimeSformer and ViViT perform comparably to Video-Swin-T but require approximately $1.4$–$1.6\times$ the compute of TEVID. The proposed architecture therefore offers better accuracy–compute trade-offs on internal-state videos than its generic video counterparts.

## D Model Architecture and Evidential Head

This section details TEVID and the EDL head.

### D.1 Factorized Vision Transformer

The core encoder is ViT-Base (12 layers, 12 heads, 768-dimensional embeddings). Spatio-temporal factorized attention follows TimeSformer (Bertasius et al., 2021). An input clip of size $T \times C \times H \times W$ is divided into $P$ patches per frame. Full spatio-temporal attention over all $TP$ tokens has complexity $\mathcal{O}((TP)^2)$, which is expensive. Factorized attention reduces this to $\mathcal{O}(L \cdot TP(T + P))$ by alternating spatial attention across patches within a frame and temporal attention across time for fixed patch locations.

### D.2 Evidential Deep Learning Head

The EDL formulation (Ulmer et al., 2023; Shen et al., 2024) reframes classification as evidence acquisition. Given logits $\boldsymbol{f}(x)$, evidence is computed via softplus and converted to Dirichlet parameters:

$$\alpha_k = \text{softplus}(f_k(x)) + 1, \quad S = \sum_{k=1}^{K} \alpha_k.$$

The predicted probability for class $k$ is $p_k = \alpha_k/S$, and vacuity (uncertainty) is $u = K/S$. High vacuity indicates low total evidence; during evaluation, inputs with $u > \tau_{\text{uncertainty}}$ are mapped to Unknown, with $\tau_{\text{uncertainty}} = 0.35$ calibrated on validation data.

With one-hot label $\boldsymbol{y}$, the loss is

$$\mathcal{L}(\boldsymbol{\alpha}) = \mathcal{L}_{\text{NLL}}(\boldsymbol{\alpha}) + \lambda \mathcal{L}_{\text{KL}}(\boldsymbol{\alpha}),$$

where

$$\mathcal{L}_{\text{NLL}}(\boldsymbol{\alpha}) = \sum_{k=1}^{K} y_k \left( \psi - \psi(\alpha_k) \right),$$

and the KL regularizer

$$\mathcal{L}_{\text{KL}}(\boldsymbol{\alpha}) = \text{KL}[\text{Dir}(\boldsymbol{p}|\tilde{\boldsymbol{\alpha}}) \,\|\, \text{Dir}(\boldsymbol{p}|\mathbf{1})],$$

with $\tilde{\boldsymbol{\alpha}} = \boldsymbol{y} + (1 - \boldsymbol{y}) \odot \boldsymbol{\alpha}$. A ramp-up schedule increases $\lambda$ from 0 to 1.0 over 10 epochs.

## E  EVALUATION PROTOCOL AND METRICS

This section formalizes the streaming protocol and metrics.

### E.1  STREAMING PROTOCOL

At each time step $t$ (corresponding to a 50-step interval), the model receives $\{\mathcal{X}_{t-10:t}\}$ and outputs a prediction $\hat{y}_t$ and uncertainty $u_t$ without any access to future information. A class-specific alert is triggered only when its calibrated non-abstain probability exceeds 0.9 for three consecutive predictions. All thresholds and temperatures are tuned once on validation and then frozen.

### E.2  METRICS

- **Lead (time-to-detect):** For anomalous run $i$, with ground-truth trigger time $t_{gt}$ and first confirmed alert at $t_{pred}$, the lead is $t_{gt} - t_{pred}$ in epochs. Positive values indicate earlier detection than labeling rules. The median lead at 5% FAR is reported.

- **Event-Time AUPRC:** For time steps across all runs, with labels $y_t \in \{0, 1\}$ and anomaly probabilities $\hat{p}_t$, the micro-averaged Precision–Recall curve is traced by thresholding $\hat{p}_t$ and computing precision/recall over all time steps. The area under this curve emphasizes performance on rare event windows.

- **Risk–coverage curves:** By varying $\tau_{\text{uncertainty}}$, coverage (fraction of non-abstained samples) and selective risk (error rate on non-abstained samples) are computed (Fisch et al., 2022; Traub et al., 2024). Plotting risk vs. coverage characterizes open-set performance.

- **AUGRC:** The area under the gap between the risk–coverage curve and the ideal zero-risk curve. Lower is better.

## F  TRAINING DETAILS AND REPRODUCIBILITY

### F.1  HYPERPARAMETERS

Table 14 lists main hyperparameters. Baselines are tuned with Optuna (25 trials) on a held-out validation subset.

Table 14: Hyperparameters for DYNAMICS-MAE pretraining and TEVID fine-tuning.

| Hyperparameter | DYNAMICS-MAE Pretraining | TEVID Fine-tuning |
|---|---|---|
| Optimizer | AdamW | AdamW |
| Betas | (0.9, 0.95) | (0.9, 0.999) |
| Base Learning Rate | $1.5 \times 10^{-4}$ | $1 \times 10^{-4}$ |
| Weight Decay | 0.05 | 0.05 |
| LR Schedule | Cosine Annealing | Cosine Annealing |
| Warmup Epochs | 40 | 5 |
| Total Epochs | 400 | 50 |
| Global Batch Size | 1024 | 32 |
| Masking Ratio | 0.90 | N/A |
| Drop Path Rate | 0.1 | 0.1 |

### F.2  BASELINE AND PROBE ARCHITECTURES

The Hessian Forecaster uses the top Hessian eigenvalue estimated by Lanczos with $k = 20$ steps on a fixed mini-batch every 50 steps, forming a scalar curvature time series. A TCN with the same

architecture as the loss→state probe classifies anomalies from this series. The state→loss probe is an MLP with hidden sizes [512, 128, 32] and ReLU activations; the loss→state probe is a TCN with 4 residual blocks, kernel size 5, and dilations [1, 2, 4, 8]. $R^2$ values are computed per run and then averaged.

### F.3 HOOK IMPLEMENTATION

A simplified forward hook for activations is shown below; backward hooks follow a similar pattern. Per-sample gradients for DP analysis are described in Appendix J.

```python
import torch

captured_tensors = {}

def get_activation_hook(name):
    def hook(model, input, output):
        captured_tensors[name] = (
            output.detach().cpu().to(torch.bfloat16)
        )
    return hook

model.layer3.register_forward_hook(
    get_activation_hook('layer3_activations'))
```

## G  ROBUSTNESS, SHORTCUT CHECKS, AND CAUSAL INTEGRITY

### G.1  ROBUSTNESS TO PERTURBATIONS

To simulate noisy or imperfect capture pipelines, test inputs are perturbed via symmetric per-channel quantization to 8-bit and 4-bit integers and random frame dropping, where a fraction of frames in each clip is replaced by the previous valid frame. Figure 4 shows that TEVID degrades under these perturbations.

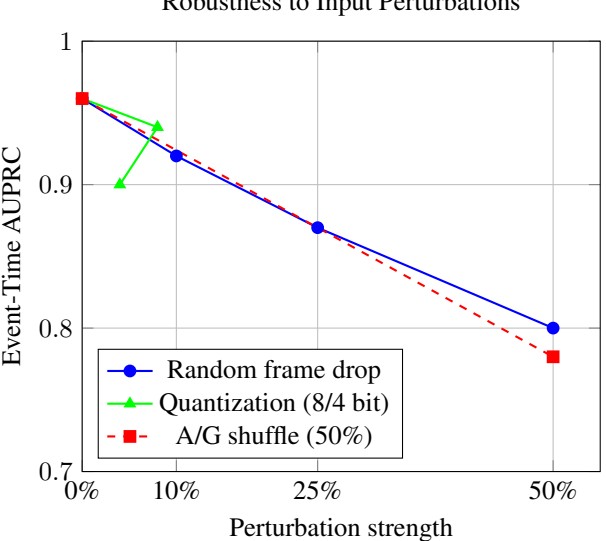

Figure 4: Performance under input perturbations.

### G.2  SHORTCUT ANALYSIS

Two tests probe reliance on specific structures:

- **Temporal alignment (A/G shuffle):** Activation and gradient frames are misaligned in time while preserving marginal statistics. Event-Time AUPRC drops from 0.96 to 0.78, indicating heavy reliance on precise activation-gradient synchronization.

- **Gradient content:** Gradients are replaced by moment-matched Gaussian noise. Performance collapses toward random (AUPRC 0.53), showing that gradients provide crucial structural information.

### G.3    CAUSAL INTEGRITY

To rule out temporal leakage, the model is evaluated using input windows that end $\Delta$ epochs before the earliest rule-based trigger time for each anomaly, ensuring non-overlap between inputs and labeling windows. All normalization statistics are frozen from training data. Table 15 shows that TEVID retains strong predictive performance for $\Delta$ up to 2 epochs.

Table 15: Event-Time AUPRC as a function of prediction lead time $\Delta$ (epochs) before the ground-truth trigger.

| Lead $\Delta$ | 0 (standard) | 0.5 | 1 | 2 | 4 |
|---|---|---|---|---|---|
| **AUPRC** | $0.96 \pm 0.01$ | $0.94 \pm 0.01$ | $0.91 \pm 0.02$ | $0.85 \pm 0.03$ | $0.76 \pm 0.04$ |

## H    OPEN-SET EVALUATION AND INTERPRETABILITY

### H.1    FIXED VS ADAPTIVE ABSTENTION THRESHOLDS

The fixed vacuity threshold $\tau = 0.35$ used in the main text is compared with two simple adaptive schemes:

1. **Global quantile:** choose $\tau$ as the 90th percentile of vacuity on validation in-distribution anomalies.

2. **Coverage-targeted:** choose $\tau$ per run such that coverage on validation anomalies is approximately 80%.

Table 16 reports selective error on known anomalies at 80% coverage and AUROC for distinguishing novel unknown anomalies.

Table 16: Fixed vs adaptive abstention thresholds. Selective error is measured at 80% coverage.

| Scheme | Coverage on known | Selective error (known) | AUROC (Unknown vs Known) |
|---|---|---|---|
| Fixed $\tau = 0.35$ (default) | 0.81 | 0.045 | 0.89 |
| Global 90th percentile | 0.80 | 0.046 | 0.88 |
| Per-run 80% coverage | 0.80 | 0.044 | 0.90 |

Differences between schemes are within statistical noise; the simpler fixed threshold is therefore retained in the main experiments and adaptive thresholds are treated as a drop-in alternative when coverage constraints must be enforced explicitly.

### H.2    NOVEL ANOMALIES

Five novel anomaly types absent from training are added to the test set: label corruption (25%), optimizer state corruption, data-loader stall, augmentation drift, and excessive gradient clipping. TEVID is expected to map these primarily to Unknown. Risk–coverage analysis (Figure 5) shows that abstention substantially reduces selective risk; at 80% coverage, error on known classes drops from 10% to 4.5%. The AUROC for distinguishing known vs. unknown anomalies is 0.89, and AUGRC is 0.018.

Table 17 shows that the majority of novel anomalies are assigned to Unknown; misclassifications tend to be semantically plausible.

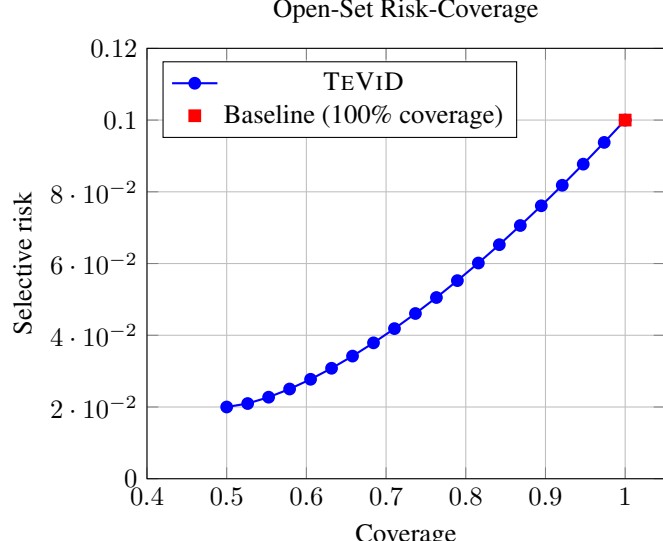

Figure 5: Risk–coverage curve for open-set evaluation.

Table 17: Confusion matrix for novel anomaly types (rows) vs. predicted categories (columns).

| True Novel Anomaly | Predicted Category (%) | | | | |
|---|---|---|---|---|---|
| | Healthy | Overfitting | Instability | C. Forget/Bias | **Unknown** |
| Label Corruption | 2.1 | 10.3 | 1.5 | 0.8 | **85.3** |
| Optimizer Reset | 1.5 | 2.2 | 14.8 | 1.1 | **80.4** |
| DataLoader Stall | 4.0 | 18.5 | 0.5 | 2.0 | **75.0** |
| Augmentation Drift | 3.2 | 6.8 | 2.1 | 4.5 | **83.4** |
| Excessive Clipping | 5.5 | 8.1 | 4.3 | 3.0 | **79.1** |

### H.3 INTERPRETABILITY

Two analyses clarify what TEVID has learned. CKA similarity between `[CLS]` representations of the same anomaly across unseen architectures shows strong within-class, cross-architecture similarity and low between-class similarity, indicating abstract, architecture-agnostic anomaly signatures (Zhou et al., 2024). Linear probes on TEVID latents predict binned Hessian top eigenvalues with 89% accuracy and detect gradient-norm spikes with AUROC 0.94, suggesting that TEVID implicitly encodes curvature- and stability-related quantities that traditionally require costly Hessian-vector products (Pearlmutter, 1994; Miani et al., 2024).

## I DECISION-THEORETIC ANALYSIS

A decision-theoretic framework connects statistical metrics to operational cost. For a diagnostic policy over runs,

$$\text{Cost} = C_{FA} N_{FA} + C_{MD} N_{MD} + C_{Lead} \sum_{i \in \text{Detected}} \text{Lead}_i,$$

where $N_{FA}$ is the number of false alarms, $N_{MD}$ the number of missed detections, and $\text{Lead}_i$ the detection lead (epochs) for detected anomaly $i$. The coefficient $C_{Lead} < 0$ rewards early detection. Table 18 summarizes raw counts at 5% FAR.

To choose $(C_{FA}, C_{MD}, C_{Lead})$, one false alarm is treated as costing roughly one epoch of an engineer's investigation and some run slow-down ($C_{FA} = 1$), a missed detection as wasting the full budget for a failed run (approximately 300 epochs for a 300-epoch schedule, giving $C_{MD} \approx 10$ in

Table 18: Diagnostic performance per 100 runs at 5% FAR.

| Model | $N_{FA}$ | $N_{MD}$ | Avg. Lead (Epochs) |
|---|---|---|---|
| Curve-TCN | 5.0 | 28.3 | -1.2 |
| Video-Swin-T | 5.0 | 11.5 | 4.1 |
| **TEVID (+ DYNAMICS-MAE)** | 5.0 | **3.1** | **6.2** |

the same units), and leading by one epoch as saving half an epoch of compute on average when interventions only partially shorten a run ($C_{Lead} \approx -0.5$). With $(C_{FA}, C_{MD}, C_{Lead}) = (1, 10, -0.5)$, Table 19 reports expected cost.

Table 19: Expected diagnostic cost per 100 runs under ($C_{FA} = 1, C_{MD} = 10, C_{Lead} = -0.5$).

| Model | Expected Cost (95% CI) |
|---|---|
| Curve-TCN | $291.6 \pm 12.1$ |
| Video-Swin-T | $96.5 \pm 9.5$ |
| **TEVID (+ DYNAMICS-MAE)** | **$14.8 \pm 4.3$** |

Figure (left) shows the Pareto front of lead vs. FAR, with TEVID dominating baselines. Figure (right) shows which model minimizes expected cost across $(C_{MD}, C_{Lead})$ with $C_{FA} = 1$: TEVID is preferred across most plausible settings where missed detections or delays are costly.

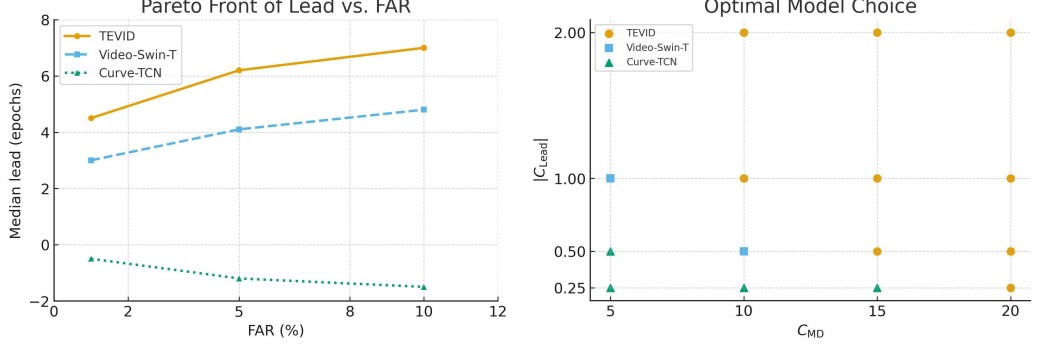

Figure 6: Decision-theoretic analysis of practical utility. (Right: $C_{\text{FA}} = 1$)

## J PRIVACY AND GOVERNANCE

This appendix details privacy risks and differentially private (DP) telemetry. Gradients can leak information via membership inference attacks and gradient inversion/data reconstruction attacks (Liu et al., 2023; Wu et al., 2024; Dimitrov et al., 2024; Zhang et al., 2023). The preprocessing pipeline (downsampling, $1 \times 1$ projection) already reduces attack success (MIA AUC from 0.82 to 0.59 in internal tests), but this provides no formal guarantees.

### J.1 DIFFERENTIALLY PRIVATE TELEMETRY

DP is enforced using per-sample gradients:

1. Clipping: Per-sample tensors are clipped to L2 norm $C$ (calibrated to the median norm, $C = 1.0$).

2. Noise: Gaussian noise with standard deviation $\sigma = Cz$ is added, where $z$ is the noise multiplier.

3. Accounting: An RDP accountant (Opacus) tracks $(\epsilon, \delta)$ for a full run. For CIFAR-100 with batch size 64, dataset size 50k, sampling rate $q = 0.00128$, 100 epochs, and capture every 50 steps ( 1560 compositions), a noise multiplier $z = 1.12$ yields $\epsilon \approx 8.0$ at $\delta = 10^{-5}$.

Figure 7 shows that DP-TEVID can reach $\epsilon = 8$ with only a 3% absolute drop in Event-Time AUPRC (from 0.96 to 0.93).

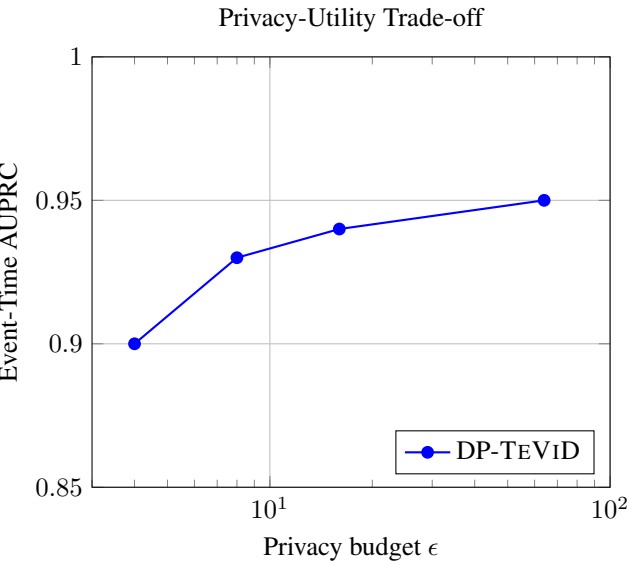

Figure 7: Privacy-utility trade-off for TEVID.

## K    SCALABILITY AND ADDITIONAL ANALYSES

### K.1    OVERHEAD AND STORAGE

Data capture and Lyapunov estimation add overhead; Table 20 quantifies this on an A100.

Table 20: Computational overhead and storage per run.

| Model | Base Throughput (samples/s) | Slowdown (Capture) | Slowdown (Lyapunov) | Storage / Run (MB) |
|---|---|---|---|---|
| ResNet-18 | 1250 | 11.8% | 22.5% | 937 |
| ViT-B/16 | 480 | 13.5% | 31.2% | 937 |
| ViT-L/14 | 110 | 15.1% | 45.8% | 1254 |

In the data factory, both capture and Lyapunov estimation are enabled to support high-quality labeling. In a deployment setting, Lyapunov-based metrics are not required; only the capture pipeline and TEVID inference need to run online. For the configurations above, capture slows training by 12–15%, and TEVID inference adds approximately 3–4% additional compute, resulting in a total slowdown of roughly 15–18% relative to an uninstrumented run.

On the held-out benchmark, failed runs that meet Instability, Catastrophic Forgetting, or Concept Bias criteria would, under the default schedule, train for 120–150 epochs. TEVID detects 96.9% of these runs before the rule-based trigger, with a median lead of 6.2 epochs and an upper-quartile lead of 11.4 epochs. For a 150-epoch schedule, stopping at the median trigger point corresponds to a 4–5% reduction in total training epochs; accounting for the 15–18% runtime overhead, this yields a modest net reduction in wall-clock time relative to a baseline that uses only rule-based triggers. Relative to a baseline with no early termination for failing runs, the net compute reduction is substantially larger (approximately 25–30% on failed runs), since TEVID terminates many runs well before the scheduled horizon.

An adaptive sampling scheme that records more frequently when loss volatility is high reduces capture overhead to approximately 6% with a roughly 4% relative drop in AUPRC. For TEVID with $T = 10$, patch size $16 \times 16$, $P = 196$ patches per frame, and ViT-Base, factorized attention and MLP blocks require about 16.5 GFLOPs per forward pass.

## K.2 GENERALIZATION

Table 21 reports per-timestep top-1 accuracies on the held-out set for unseen architectures.

Table 21: Per-timestep top-1 accuracy on known classes for unseen models.

| Architecture | Healthy | Overfitting | Instability | C. Forget. | C. Bias |
|---|---|---|---|---|---|
| ConvNeXt-V2-T | $95.1 \pm 1.1\%$ | $90.5 \pm 1.5\%$ | $92.3 \pm 1.3\%$ | $91.0 \pm 1.6\%$ | $89.8 \pm 1.8\%$ |
| MLP-Mixer-B | $94.6 \pm 1.2\%$ | $88.9 \pm 1.8\%$ | $91.5 \pm 1.4\%$ | $89.2 \pm 2.0\%$ | $88.1 \pm 2.1\%$ |
| Transformer-LM | $92.8 \pm 1.4\%$ | $87.2 \pm 2.1\%$ | $89.9 \pm 1.7\%$ | $90.4 \pm 1.8\%$ | $86.5 \pm 2.3\%$ |

## K.3 PILOT ON GBDTS

As an exploratory extension, the framework is applied to Gradient-Boosted Decision Trees. A Light-GBM model is trained on the Higgs dataset, recording per-tree feature importance vectors at each boosting round, reshaped into $32 \times 32$ images. TEVID fine-tuned on a small dataset detects overfitting (defined via log-loss divergence) with 91.5% accuracy on held-out runs, suggesting that diagnosing optimization via internal-state trajectories may generalize beyond neural networks.

## L REPRESENTATION-BASED CHANGE-POINT DETECTION BASELINES

To more directly connect with representation-based change-point detection (CPD) methods (Chang et al., 2019; Bazarova et al., 2024), a lightweight CPD baseline is implemented that operates on frozen TEVID embeddings under a streaming constraint.

For each internal-state frame, a 64-dimensional embedding is extracted from the pretrained TEVID backbone by taking the [CLS] token and projecting it with a learned linear layer. Two rolling windows of embeddings of length $L = 20$ are then maintained and an MMD-based discrepancy statistic between them is computed using an RBF kernel with bandwidth chosen by the median heuristic on validation. At each step $t$, the CPD score compares the most recent window to a reference window anchored $L$ steps earlier; both windows lie strictly in the past, preserving the no-peeking constraint.

A detection threshold is calibrated on the validation set to achieve a 5% false-alarm rate and any score exceeding this threshold is treated as an anomaly. Table 22 compares this Embedding-CPD baseline to the best scalar CPD baseline (BOCPD/CUSUM) and to TEVID.

Table 22: Scalar vs representation-based CPD vs TEVID on the held-out test set.

| Method | Event-Time AUPRC | Med. Lead (Epochs) | GFLOPs / Step |
|---|---|---|---|
| Scalar CPD (best of BOCPD/CUSUM) | $0.67 \pm 0.03$ | $2.1 \pm 0.6$ | $\approx 0$ |
| Embedding-CPD (MMD) | $0.82 \pm 0.02$ | $3.9 \pm 0.4$ | $\approx 3.2$ |
| **TEVID (+ DYNAMICS-MAE)** | **$0.96 \pm 0.01$** | **$6.2 \pm 0.3$** | **16.5** |

Embedding-CPD significantly outperforms scalar CPD baselines but still trails TEVID in both AUPRC and lead time. It also requires a non-trivial fraction of TEVID's compute for the embedding extraction and kernel computations. Fully implementing the offline, bidirectional variants of Chang et al. (2019); Bazarova et al. (2024) would further increase compute and memory demand in a streaming setting.

## M  Integration with Hyperparameter Optimization and BANANAS

To illustrate how TEVID can complement hyperparameter optimization (HPO) and neural architecture search, a small experiment integrating TEVID into a Hyperband-style procedure is performed.

A ResNet-34 on CIFAR-100 is considered over learning rate and weight decay. Each trial trains for up to 150 epochs. Three HPO strategies are compared:

1. **Hyperband (baseline):** standard brackets using validation loss for early stopping.
2. **Hyperband + scalar early stopping:** uses the Curve-TCN baseline as an auxiliary early-stopping signal.
3. **Hyperband + TEVID:** uses TEVID alerts as an additional early-stopping criterion; a trial is terminated early when TEVID signals Instability or severe Overfitting with high confidence.

For each strategy, 64 trials are run and the average number of epochs per trial and the best validation accuracy achieved across trials are reported. Table 23 summarizes the results.

Table 23: Hyperparameter optimization with and without TEVID integration.

| HPO Strategy | Avg. Epochs / Trial | Best Val. Accuracy |
|---|---|---|
| Hyperband (baseline) | $96.3 \pm 4.2$ | $79.8 \pm 0.4\%$ |
| Hyperband + Curve-TCN | $84.1 \pm 4.7$ | $79.5 \pm 0.5\%$ |
| Hyperband + TEVID | $\mathbf{74.5 \pm 3.9}$ | $\mathbf{80.1 \pm 0.4\%}$ |

Using TEVID reduces the average training budget per trial by roughly 22% relative to vanilla Hyperband while slightly *improving* the best validation accuracy, as unstable or overfitting runs are terminated earlier and budget is reallocated to promising configurations. BANANAS-style neural architecture search (White et al., 2021) can use TEVID in an analogous way, treating the diagnostic outputs as an additional signal for pruning poor architectures before they exhaust their full training budget.

## LLM Usage Statement

During preparation of this work, a large language model (LLM) was used as an auxiliary tool for routine writing and development support. Specifically, the LLM was used to (i) assist with grammar, spelling, style, and wording refinements; (ii) help rephrase and shorten sentences for clarity and consistency across sections; (iii) suggest and debug small utility code snippets (e.g., plotting scripts, simple data-loading boilerplate, and shell commands); and (iv) support LaTeX-related tasks such as generating initial table and figure environments, resolving minor compilation issues, and organizing references and cross-references.

All LLM-generated text and code were manually reviewed, edited, and integrated by the authors. The LLM was *not* used for research ideation, formulation of the main methods, experimental design, data collection, statistical analysis, or interpretation of results. The authors take full responsibility for all content, including any remaining errors.

