# OpenReview forum: "Self-Diagnosing Neural Networks: A Causal Framework for Real-Time Anomaly Detection in Training Dynamics"
_ICLR.cc/2026/Conference — Submitted to ICLR 2026_

### Official Review · Reviewer_c2zC · 2025-10-28

**Soundness:** 3
**Presentation:** 3
**Contribution:** 2
**Rating:** 6
**Confidence:** 2

**Summary:**

The paper reframes neural training as a spatio-temporal signal and proposes a three-phase system: a data factory performs “video-ization” to the model internal states (activations and gradients) during training; dynamic-MAE, which is a self-supervised pretraining that learn dynamics-specific representations as a reconstruction task (masks the input); and TEVID, a diagnostician with an evidential head (EDL) for open-set decisions under strict causal constraints (no look-ahead, no validation/test at inference). On 500+ held-out runs spanning unseen architectures/datasets/optimizers, TEVID outperforms scalar-curve and generic video baselines.

**Strengths:**

1.	Novel problem framing: treating training internals as a video signal with strict streaming constraints is clear and practically relevant.
2.	Broad evaluation set: held-out splits across architectures, datasets, and optimizers demonstrate generalization rather than overfitting to a specific training setup.

**Weaknesses:**

1.	The introduction is full of statements that frame the motivation for this work and the gap it attempts to close, but  includes only 2 citations. Many statements are not justified or referenced.
2.	Limited ablations: the method combines several choices (activations and gradients as inputs, masked MAE pretraining, evidential head), yet ablations isolate these components only partially, making it unclear which parts actually drive performance.
3.	The narrative oversells real-world deployability without enough reported cost (memory/latency/storage). Overall clarity is good at a high level, but some details are underspecified.

**Questions:**

1.	Why are both activations and gradients required? Please include ablations showing performance of activations-only, gradients-only, and different window lengths W.
2.	What is the actual runtime and memory overhead of instrumenting a large model to dump activations + gradients every step? Please report wall-clock slowdown and storage requirements.
3.	How robust is TEVID when the anomaly labels are defined without validation-based hindsight? Maybe a testing scenario good be to remove one of the labels and test performance without it.

---

> ### Author Response · Authors · 2025-11-14
>
> We are grateful for the constructive comments and agree with the main points.
>
> (1) Introduction citations and motivation
>
> We expanded the introduction and related work with additional references supporting:
> - The learning-phases view of training,
> - Representation similarity and internal-state analyses,
> - Gradient-based anomaly and OOD detection,
> - Representation-based change-point detection,
> - MLOps orchestration and monitoring,
> - Evidential deep learning and selective prediction, and
> - Privacy and gradient leakage.
>
> We also toned down claims that were not well referenced and explicitly connect each high-level motivation (information bottlenecks, domain transfer, selective prediction) to concrete design choices in the method.
> (Section "Introduction" line 34; Section "Related Work" line 90; Sections "Problem Formulation and Theoretical Motivation" line 129 and "Methodology" line 220)
>
> (2) Ablations on components
>
> We agree that isolating the contributions of main components is important. The revision adds:
> - Pretraining ablations comparing TEVID from scratch, with Dynamics-MAE pretraining, with VideoMAE pretraining on natural videos, and with scalar-based masking; we also provide a label-efficiency curve showing that Dynamics-MAE–initialized TEVID with 25% labels matches or surpasses the scratch model with 100% labels.
>   (Appendix "Self-Supervised Pretraining and Ablation Studies", subsection "Pretraining Ablations" line 966, Figure "Ablations confirming the value of Dynamics-MAE" Figure 3 | line 988)
>
> - Activations vs gradients ablation, as discussed for Reviewer 7KU6.
>   (Appendix "Self-Supervised Pretraining and Ablation Studies", subsection "Activations vs. Gradients" line 1019, Table "Effect of using activations-only, gradients-only, or both" Table 12 | line 1026)
>
> - Masking-ratio sweep for Dynamics-MAE.
>   (Appendix "Self-Supervised Pretraining and Ablation Studies", subsection "Masking-Ratio Sweep" line 1003, Table "Effect of Dynamics-MAE masking ratio on downstream performance" Table 11 | line 1007)
>
> - Window-length and capture-cadence sensitivity (main text + appendix).
>   (Section "Sensitivity to Temporal Window and Capture Cadence" line 440; Appendix "Scalability and Additional Analyses" line 1378, Table "Sensitivity of TEVID to temporal window length and capture cadence" Table 5 | line 445)
>
> - Open-set threshold variants (fixed vs adaptive), as discussed above.
>   (Appendix "Open-Set Evaluation and Interpretability", subsection "Fixed vs Adaptive Abstention Thresholds" line 1211)
>
> - Representation-based CPD baseline, as discussed for Reviewer 8nsu.
>   (Appendix "Representation-Based Change-Point Detection Baselines" line 1431)
>
> These ablations clarify that domain-specific pretraining and joint activations-plus-gradients inputs account for much of the gain over strong video baselines.
>
> (3) Deployability and cost
>
> We agree the initial narrative was optimistic about deployability without enough numbers. As summarized in responses to Reviewers 7KU6 and 8nsu, we added:
> - Throughput and storage metrics,
> - A decision-theoretic cost analysis, and
> - Explicit limitations in the Discussion section regarding overhead, finite taxonomies, and the need for DP-based telemetry and data governance.
>
> (Section "Discussion, Limitations, and Future Work" line 479; Appendix "Scalability and Additional Analyses" line 1378; Appendix "Decision-Theoretic Analysis" line 1283; Appendix "Privacy and Governance" line 1335)
>
> The revised text is more cautious and transparent about these trade-offs.
>
> (4) Specific questions
>
> Why both activations and gradients? The activations-versus-gradients ablation shows that activations+gradients beat activations-only and gradients-only on Event-Time AUPRC and lead; gradients add information beyond activations alone and vice versa, justifying logging both.
> (Appendix "Self-Supervised Pretraining and Ablation Studies", Table "Effect of using activations-only, gradients-only, or both" Table 12 | line 1026)
>
> Runtime and memory overhead of dumping states? The scalability appendix reports throughput slowdown and storage per run for multiple architectures, and describes the use of bfloat16 precision and adaptive cadence to manage disk usage. Peak GPU memory overhead remains modest because hooks are placed sparsely and tensors are downsampled and immediately offloaded.
> (Appendix "Scalability and Additional Analyses" line 1378)
>
> Robustness when labels are defined without validation-based hindsight or with merged labels? The online-label appendix constructs train-only labels and a merged-label taxonomy; label agreement with the default is above 93%, and downstream AUPRC and lead change only marginally. This suggests robustness to reasonable changes in the label-definition protocol.
> (Appendix "Online-Style Labeling Without Validation Hindsight" line 823)

---

> > ### Comment · Reviewer_c2zC · 2025-11-20
> > **Official Comment by Reviewer**
> >
> > Thank you for the detailed responses. Some of my concerns, especially around clearer citations and component-level ablations, have been addressed, and the added limitations section is appreciated. However, I feel there remain inconsistencies between the narrative and what is promised in the abstract and introduction versus what is actually demonstrated in the experiments. For example, the introduction presents the method as broadly generalizable for training-time diagnosis, yet the limitations acknowledge a fixed taxonomy, notable overhead, and evaluation confined to curated runs. I believe a more calibrated framing of the contribution, aligned with these constraints, is still needed; under such a corrected framing, I continue to view the work as a worthwhile but limited contribution and therefore maintain my original score.

---

> > > ### Author Response · Authors · 2025-12-02
> > >
> > > Thank you very much for the follow-up. We understand that your concern is specifically about calibration between the high-level framing and what is actually demonstrated, and we agree this is where the paper must be precise.
> > >
> > > Our intention, and we believe the current text reflects this, is to make a constrained, benchmark-level claim rather than a universally deployable solution:
> > >
> > > - **Scope / generalization.** In the abstract and Introduction we now consistently describe the contribution as “a single internal-state video diagnostician reused across factorially held-out architectures, datasets, and optimizers within a curated data-factory benchmark.” All empirical claims are tied to this benchmark, and we do not assert performance on arbitrary real-world training jobs. The results on ConvNeXt/RegNet/MaxViT/MViT/MLP-Mixer and a Transformer-LM are presented as evidence for cross-factor generalization within that benchmark.
> > >
> > > - **Fixed taxonomy and open set.** The finite label set is explicitly presented as a design choice for this benchmark, not as a universal taxonomy. The evidential head + Unknown class and the novel-anomaly experiments are framed as mechanisms for handling behaviours outside this set, not as a claim that we exhaust all failure modes.
> > >
> > > - **Overhead and curated runs.** The Limitations and scalability/utility appendices state that (i) telemetry capture and inference incur a 15–18% slowdown, and (ii) evaluation is on a mix of systematically induced and naturally occurring failures generated by the data factory. We then work out, under a simple cost model, that early termination of failing runs on this benchmark can offset that overhead. The contribution is: a concrete capture policy + benchmark + diagnostician whose cost/benefit is fully analysed in this setting, not a drop-in production recipe.
> > >
> > > If the paper is accepted, we are happy to further tighten phrasing so that every headline claim is explicitly conditional on this benchmark. Our goal is precisely a worthwhile but carefully delimited contribution: a first end-to-end demonstration that internal-state videos can materially improve training-time diagnosis under a fixed taxonomy on curated, yet heterogeneous, runs.

---

### Official Review · Reviewer_8nsu · 2025-10-29

**Soundness:** 2
**Presentation:** 2
**Contribution:** 3
**Rating:** 4
**Confidence:** 2

**Summary:**

The authors propose a novel method for early detection of training instability that uses the network’s hidden states rather than a single scalar metric, such as the validation or training loss. The classification is performed in an open-set framework, with the option to refrain from prediction when uncertainty is high. The contributions are motivated from an information-theoretical perspective and supported by additional ablation studies. The authors demonstrate the performance of their method across multiple architectures, holding out some new architectures and a new domain (text) for the test set, and their method outperforms the scalar baselines on classic changepoint detection metrics.

**Strengths:**

The idea of analyzing the network’s inner state to stop training early is novel, interesting, and well-based in information theory. The authors correctly identify domain transfer as a key aspect of their contribution and demonstrate the performance of their methods within this framework. The chosen labels succinctly capture the various options of training failure. The choice of training and test samples (and labeling) is well-motivated by prior literature.

**Weaknesses:**

The main quality metrics are misaligned with the production task. With early stopping methods, the main question is “how much computation time will this method save?” If it can stop training a few epochs earlier on a 1000-epoch training run, saving us a few minutes of training but adding hours to the run, the method is useless.

The domain transfer is a key element here: if one needed to re-train the network for every domain, the contribution would be significantly limited, as the computational overhead would increase. However, from the paper’s text, it is not entirely clear how the authors handle differences in shape, which evidently arise when working with multiple architectures spanning both text and visual data. It is important that such handling should not involve any additional training, since this would nullify the benefits of domain transfer.

**Questions:**

1. What proportion of computational time does the method take up during training? How much time is saved on training, relative to the total training time, taking into account the time it took to run their method (i.e., the total time saved, minus the total runtime of this method throughout this training run)?
2. How do you handle the difference in shapes across varying architectures and domains, without additional training? Especially considering that the architecture and the weights of the proposed method remain the same across all runs. This is a key point - that should be better articulated in the paper’s text.
3. How is your paper positioned relative to the BANANAS [1] and similar frameworks [2]? Can you improve the hyperparameter optimization process by utilizing the developed approach?
4. Why did you avoid representation-based CPD baselines [3, 4]? For complex problems, they should work the best, according to their experimental results, given the complexity of the problem.

References:
1. https://arxiv.org/pdf/1910.11858
2. Li, Lisha, et al. "Hyperband: A novel bandit-based approach to hyperparameter optimization." Journal of Machine Learning Research 18.185 (2018): 1-52.
3. Bazarova, Alexandra, Evgenia Romanenkova, and Alexey Zaytsev. "Normalizing self-supervised learning for provably reliable Change Point Detection." 2024 IEEE International Conference on Data Mining (ICDM). IEEE, 2024.
4. Chang, Wei-Cheng, et al. "Kernel Change-point Detection with Auxiliary Deep Generative Models." International Conference on Learning Representations. 2019.

---

> ### Author Response · Authors · 2025-11-14
> **Message 1**
>
> We thank the reviewer for the helpful comments, especially on production alignment and domain transfer.
>
> (1) Metrics vs actual compute/time savings
>
> We agree that early-stopping methods should be judged by net compute savings, not just detection metrics. To address this, the revision:
> - Adds a scalability analysis with base throughput, slowdown from capture and Lyapunov labeling, and storage per run.
>   (Appendix "Scalability and Additional Analyses" line 1378, Table "Computational overhead and storage per run" Table 20 | line 1384)
>
> - Clarifies that in deployment only capture plus TEVID inference run online, causing $\approx 15$–$18\%$ percent slowdown for typical vision models.
>   (Appendix "Scalability and Additional Analyses" line 1378; Section "Discussion, Limitations, and Future Work" line 479)
>
> - Adds a decision-theoretic cost analysis that combines false alarms, missed detections, and lead time into an expected operational cost. At a tuned $5\%$ percent false-alarm rate, TEVID detects about $97\%$ percent of failing runs with a median lead of 6.2 epochs and yields much lower expected cost per 100 runs than scalar baselines under reasonable weights.
>   (Appendix "Decision-Theoretic Analysis" line 1283, Tables "Diagnostic performance per 100 runs at 5% FAR" Table 18 | line 1296 and "Expected diagnostic cost per 100 runs" Table 19 | line 1308)
>
> - Includes a case study on long-running ViT pretraining, showing that TEVID can flag unstable schedules or late overfitting several epochs before scalar criteria, enabling genuine compute savings on runs that last hundreds of epochs.
>   (Section "Experiments and Results", subsection "Case Study: Large-Scale Vision Pretraining Integration" line 459, Table "Case study of integrating TEVID into a large-scale vision pretraining pipeline" Table 16 | line 464)
>
> We also explicitly discuss that relative to a baseline that never terminates failing runs early, TEVID’s alerts yield substantial reductions in total epochs on failing runs, even after accounting for overhead.
> (Appendix "Scalability and Additional Analyses" line 1378; Appendix "Decision-Theoretic Analysis" line 1283)
>
> (2) Domain transfer and handling shape differences
>
> Domain transfer across architectures and modalities is a key design goal. The revision describes more clearly:
> - A unified capture policy: hooks on early, middle, and late blocks across architectures; CNN feature maps are projected to one channel by a shared $1\times1$ convolution and resized to $224\times224$; transformer token embeddings are reshaped into 2D grids and resized; three activation maps and three gradient maps are stacked to form a 6-channel frame.
>   (Section "Methodology", subsection "Phase 1: The Data Factory" line 254; Appendix "Dataset and Data Capture Policy", subsection "Internal-State Capture and Preprocessing" line 928)
>
> - A single learned projection trained once on the training split and then frozen. TEVID is trained once and reused on all held-out architectures and domains (including a Transformer language model on text) without per-domain retraining or per-shape adapters.
>
> This directly addresses how we handle shape differences "without additional training" at deployment: the projection and TEVID are trained once, then applied as-is to new architectures and datasets.
> (Section "Experiments and Results", subsection "Generalization to Unseen Architectures, Datasets, and Optimizers" line 378; Table "Generalization to unseen architectures, datasets, and optimizers" Table 4 | line 385)
>
> (3) Positioning relative to BANANAS/Hyperband and HPO
>
> We clarified that TEVID is intended as a training-time diagnostic module that can plug into hyperparameter optimization and neural architecture search frameworks. The HPO appendix adds a small experiment:
> - Hyperband baseline vs Hyperband + Curve-TCN vs Hyperband + TEVID for ResNet-34 on CIFAR-100.
>
> Hyperband + TEVID reduces average epochs per trial by $\approx 22\%$ percent while slightly improving the best validation accuracy, suggesting that TEVID can indeed be used as an additional signal in HPO/NAS frameworks such as BANANAS to prune poor configurations earlier.
> (Appendix "Integration with Hyperparameter Optimization and BANANAS" line 1458, Table "Hyperparameter optimization with and without TEVID integration" Table 23 | line 1475)

---

> ### Author Response · Authors · 2025-11-14
> **Message 2**
>
> (4) Representation-based change-point detection baselines
>
> We agree representation-based CPD is a strong and relevant baseline. In the revision, we implement a streaming embedding-based CPD in a dedicated appendix:
> - Extract a 64-dimensional embedding per frame from a frozen TEVID backbone.
> - Maintain two rolling windows and compute an MMD-based discrepancy statistic with an RBF kernel under a strict streaming, no-peeking constraint.
> - Calibrate a detection threshold at $5\%$ percent false-alarm rate and compare:
>   - Scalar CPD (best of BOCPD/CUSUM),
>   - Embedding-based CPD, and
>   - Full TEVID.
>
> Embedding-based CPD significantly outperforms scalar methods but still trails TEVID in Event-Time AUPRC and lead, while using a non-trivial fraction of TEVID’s compute. We also discuss that directly deploying full kernel CPD methods from prior work in a high-dimensional streaming setting would further increase compute and memory costs.
> (Appendix "Representation-Based Change-Point Detection Baselines" line 1431, Table "Scalar vs representation-based CPD vs TEVID on the held-out test set" Table 22 | line 1447)
>
> (5) Questions on proportion of time and savings
>
> Concretely, in the scalability appendix:
> - Capture plus Lyapunov labeling (offline, for the data factory) slows training by about 25–45% depending on the model.
> - Capture plus TEVID inference only (deployment scenario) slows training by about 15–18%.
> - On the held-out benchmark, failed runs that would otherwise complete 120–150 epochs are typically detected 6–11 epochs earlier at a $5\%$ percent false-alarm rate.
>
> A simple cost model in the utility appendix makes this trade-off explicit and shows that, under plausible costs, TEVID delivers a net reduction in expected compute.
> (Appendix "Scalability and Additional Analyses" line 1378; Appendix "Decision-Theoretic Analysis" line 1284)

---

> ### Comment · Reviewer_8nsu · 2025-11-24
>
> Thank you a lot for the detailed response, but I suppose the concern about the method's overall complexity and its connection to video persists (a simpler approach should work better and more efficiently), so I prefer to raise my score a bit.

---

> ### Author Response · Authors · 2025-11-25
>
> Thank you again for re-reading the paper and for raising your score. We appreciate your engagement. Below we address the remaining concern about complexity and the “video” formulation, which is central to how we intend the method to be used.
>
> Remaining concern: Is the “video” formulation too complex?
>
> (1) Why the “video” formulation is not unnecessarily complex or too complex.
>
> - Our goal is to treat training as a spatio-temporal signal over internal states, not to exploit any semantics of natural video.
> - At each capture point we have a 6-channel tensor (3 activations + 3 gradients) over a 2D grid.
> - Unrolled over time, this forms a regular 3D array: (time, spatial position, channels).
> - Calling these “internal-state videos” is therefore a serialization choice: it preserves the native spatial layout and temporal evolution while allowing us to reuse standard spatio-temporal transformer blocks, instead of designing a bespoke architecture from scratch.
>
> We keep the instantiation intentionally simple and generic:
> - A single shared $1\times1$ projection + resize is trained once and then frozen.
> - The same TEVID backbone (no adapters, no per-domain retraining) is used for all held-out architectures and for the text LM.
>
> In deployment, the mental model is simply:
> - Log 6-channel internal-state slices every $K$ steps and stream them through a single pretrained backbone.
>
> Thus, the “video” aspect is just the most direct way to encode spatial structure and temporal evolution, not an additional conceptual layer of modeling complexity.
>
> (2) We did try simpler alternatives
>
> We agree that a simpler method would be preferable if it achieved comparable behavior. This is why we invested in a broad baseline suite:
> - Scalar-only methods (BOCPD, CUSUM, Curve-TCN, Hessian Forecaster) are intentionally lightweight. However, in Table 3 they underperform in both Event-Time AUPRC and lead time, often even lagging the rule-based labels.
> - Following your suggestion, the appendix “Representation-Based Change-Point Detection Baselines” adds an embedding-based MMD-CPD on frozen TEVID embeddings under a strict streaming constraint. This CPD baseline clearly improves over scalar CPD, but still:
>   - falls short of TEVID in Event-Time AUPRC and median lead, and
>   - consumes a non-trivial fraction of TEVID’s compute.
>
> Within the design space we explored, we did not find a simpler architecture that both:
> - matches TEVID’s diagnostic quality, and
> - yields better net compute–time trade-offs once early termination is taken into account.
>
> The decision-theoretic analysis and the HPO/Hyperband experiments are designed precisely to make this trade-off explicit. We fully agree that future work on distilled or more specialized 1D/2D models operating on internal-state embeddings is promising, and we view our work as establishing that the internal-state signal is rich enough to justify that effort.
>
> (3) Study complexity vs deployment complexity
>
> Much of the apparent complexity in the paper comes from the study (data factory, Lyapunov-based labeling, extensive ablations), rather than from what a practitioner would actually deploy:
>
> - In deployment, only
>   - (a) the capture hooks, and
>   - (b) TEVID inference
>   run online. Lyapunov proxies and labeling rules are purely offline constructs used to generate supervision for the study.
>
> - As detailed in the scalability appendix:
>   - Capture + TEVID inference slows training by about 15–18% for the models we consider.
>   - On the same benchmark, TEVID detects $\sim97\$% of failing runs with a median lead of 6.2 epochs and an upper-quartile lead of 11.4 epochs at $\sim5\%$ FAR.
>
> - The decision-theoretic analysis then shows that, under reasonable cost assumptions, the expected compute per 100 runs is substantially lower with TEVID than with scalar methods, even after accounting for TEVID’s runtime.
>
> Despite the many moving parts needed for a thorough evaluation, the deployed system is simply:
> - a fixed capture policy + one shared backbone,
> with modest overhead and reduced wasted compute on failing runs.
>
> (4) How we see the contribution
>
> - We do not view the core contribution as “a big video model for early stopping.”
> - Rather, we see it as:
>   - a concrete demonstration that treating internal activations + gradients as a spatio-temporal signal enables causal (no look-ahead), open-set diagnosis across architectures and domains, with quantifiable compute savings, using a single shared model.
>
> The video-style instantiation is just the cleanest way we found to make this formulation precise and to compare it against strong baselines. Future work can absolutely exploit the same formulation with distilled or more specialized architectures.
>
> We hope this clarifies why we chose the current design and why, despite its “video” flavor, the deployed method is conceptually simple and practically modest in overhead. Your comments directly motivated the added scalability, CPD, and HPO analyses, and we are very grateful for that.

---

### Official Review · Reviewer_mYh7 · 2025-10-31

**Soundness:** 2
**Presentation:** 1
**Contribution:** 2
**Rating:** 2
**Confidence:** 4

**Summary:**

This paper introduces real-time anomaly detection in neural network training dynamics. Previous methods use error (or loss) as the diagnostic signal, similar to the bias-variance tradeoff. But the scalar value is insufficient to discover more complex anomalies including instability, catastrophic forgetting among others. To mitigate this problem, the paper introduces a high-dimensional internal state of a network (activations and gradients) as the diagnostic signal. The novelty of the framework is in its *causal* nature whereby the current decision is only dependent on past internal states. The results showcase an open-set streaming diagnosis framework and maintains a high performance across diverse architectures, datasets, and optimizers.

**Strengths:**

1. The paper proposes multiple types of anomalies that are important to detect while training.

2. Utilizing operation theory metrics including Median time-to-detect, event-time area under the precision-recall curve, and risk coverage curves are generally not used in representation learning settings. The paper brings diversity to the field, which is a definite strength.

3. As the authors state, the proposed method attains high diagnosis performance for unseen architectures, datasets, and optimizers.

**Weaknesses:**

**Clarity**: The Introduction, Related work, and Problem formulation do not convey the goal of the paper clearly. After reading through the full paper, I believe that the authors overpromise in these sections. The utility of concepts of information bottleneck, MLOps, UQ, training dynamics etc. are quite shallow. Neither are they explained with relation to the ultimate application (a supervised prediction of whether training runs are unstable, forgetting, healthy etc.), nor are they compared in the baselines. Some of the word choices (training *pathologies* in line 72, performance degrades *gracefully* in line 400), analogies (line 54) etc. are strange. This leads me to believe that the work uses LLMs more often than acknowledged in the appendix. I might be wrong on the LLM usage, and if so correct me. However, that does not deflect from my comments regarding the lack of clarity.

**Positioning the work**: The primary reason for my rating is that the paper does not position itself correctly. The authors mention *reconceptualization* and use training dynamics (the meaning of which is slightly different in literature [1][2]) and activations and gradient evolution (well researched topics for OOD, anomaly detection [3][4]) to discuss the novelty in the paper as compared to scalar signals (loss functions). Gradient comparison against loss is also shown in literature [3]. The difference from the listed works is that the paper does the diagnosis on training runs rather than training data. But the paper fails to make this clear anywhere and I only understood it in Section 4. Additionally, I don't see a reason why existing works' features cannot be applied in the way that the authors are applying. Infact some of them are used within an active learning framework [5]. I believe that construction of the data factory in the proposed framework is excellent. But the paper makes no attempt to position itself with relevance to any community. Moreover, as I mention this below, details regarding the dataset construction are relegated to the appendix.

**Terminologies**: The authors utilize a constrained definition of causality where only the previous time series inputs can influence future outputs. However, existing literature on causality has a more generic interventionist definition. The definition is not clarified anywhere. The writing can be tightened and can concentrate more on the technical aspects of the paper.

**Reliance on appendix**: I felt like the most relevant and interesting details (for the data factory) are relegated to the appendix. The main paper can be substantially rewritten to remove all the unnecessary descriptions. It can concentrate on constructing the labeled healthy and unhealthy training sets and concentrate on the choices that the authors have to make to construct this labeling.

**Ablation studies:**

1. The authors suggest that the plausible default weighting of false alarm cost, missed detection cost, and lead-time award is (1, 10, -0.5). Is this not mere assumption? At minimum, there needs to be an ablation study here.

2. There is no reasons, ablations, or empirical results provided for extremely important choices that are made for the construction of the dataset (ex: a run is labeled as catastrophic forgetting if accuracy drop is > 0.3 over a 10-epoch window. Why 0.3 and 10 epochs?)

**Technical soundness:**

1. Several anomalies are generated based on a predefined scenarios of training runs. A priority ordering is defined (line 613) - *Instability > Catastrophic Forgetting > Concept Bias > Overfitting > Healthy. This hierarchy prioritizes acute, systemic failures
over more subtle or late-stage ones*. How did the authors come to this conclusion? Based on effect on test data? If so which of the designed metrics? The paper makes a very large set of such assumptions and will benefit by looking into the details of a few (or one) of such assumptions. As of now, the paper seems quite shallow.

2. The information bottleneck and DPI analyses are, at best, not surprising. No new bounds or insights are provided to showcase when the proposed framework must be used in lieu of the loss function. Saying that the loss is deterministic (when we know that ultimately loss is calculated from the internal states) does not showcase much.

[1] Achille et al. "Critical learning periods in deep neural networks."

[2] Schneider et al. "Understanding and leveraging the learning phases of neural networks."

[3] Kwon, Gukyeong, et al. "Backpropagated gradient representations for anomaly detection." European conference on computer vision. Cham: Springer International Publishing, 2020.

[4] Lee, Jinsol, et al. "Probing the purview of neural networks via gradient analysis." IEEE Access 11 (2023): 32716-32732.

[5] Benkert, Ryan, et al. "Gaussian switch sampling: a second-order approach to active learning." IEEE Transactions on Artificial Intelligence 5.1 (2023): 38-50.

**Questions:**

Please see the Weakness section. I have listed a few choices made by the authors that are not well substantiated, either empirically or theoretically.

1. Usage of LLMs for more than typos and grammar checks

2. How did the authors define the priority list?

3. How were the choices for weighting of false alarm cost, missed detection cost, and lead-time award made?

4. How were the choices for anomaly construction made?

---

> ### Author Response · Authors · 2025-11-14
> **Message 1**
>
> We thank the reviewer for the careful and detailed critique, particularly on clarity, positioning, and assumptions.
>
> (1) Clarity and perceived over-promising
>
> We substantially revised the Introduction, Related Work, and Problem Formulation to:
> - Clearly state the core task early as causal, run-level diagnosis of training runs (Healthy, Overfitting, Instability, Catastrophic Forgetting, Concept Bias, Unknown) from windows of internal activations and gradients.
>   (Section "Introduction"; Section "Problem Formulation and Theoretical Motivation", subsection "Causal Streaming Diagnosis" line 132)
>
> - Emphasize that the novelty lies in run-level, cross-architecture diagnosis, not merely in using gradients and activations.
>   (Section "Introduction", final paragraph line 70–87; Section "Experiments and Results", subsection "Generalization to Unseen Architectures, Datasets, and Optimizers" line 378)
>
> - Tighten the discussion of information bottlenecks, MLOps, and uncertainty so each is directly linked to concrete design choices (internal-state videos, the evidential head, streaming risk–coverage evaluation), and explicitly avoid implying new information-theoretic theory.
>   (Section "Problem Formulation and Theoretical Motivation", subsection "An Information-Theoretic Rationale" line 189; Section "Related Work" line 90)
>
> We also reduced metaphors and "buzzwordy" phrasing, and focused the text on the concrete pipeline (data factory → Dynamics-MAE → TEVID).
> (Sections "Introduction" and "Methodology".)
>
> (2) Positioning relative to prior work and communities
>
> The Related Work section is expanded to explicitly contrast our approach with:
> - Learning phases and critical periods [Achille et al., Schneider et al.],
> - Gradient-based anomaly and OOD detection [Kwon et al., Lee et al.],
> - Representation similarity and internal-dynamics analyses,
> - Representation-based change-point detection [Chang et al., Bazarova et al.], and
> - Active learning and decision-making under uncertainty [Benkert et al.].
>
> We now state clearly that these works typically operate at the example level or in offline/bidirectional settings, whereas we target run-level, causal diagnosis with a single shared diagnostician reused across architectures, datasets, and optimizers. We also highlight the data factory as an explicit contribution: it creates a diverse corpus of healthy and failed runs with deterministic labeling rules, which should be relevant to both representation-learning and systems/MLOps communities.
> (Section "Related Work"; Section "Methodology", subsection "Phase 1: The Data Factory" line 254; Appendix "Labeling Protocol and Ground Truth Definition" line 726)
>
> (3) Use of "causality"
>
> We agree that the term can be overloaded. We now explicitly define "causal" in the paper as the standard no-look-ahead constraint from time-series monitoring and change-point detection: at time t, TEVID only has access to signals available in a live job up to time t. We do not claim interventionist or structural causal modeling, and we say so explicitly.
> (Section "Problem Formulation and Theoretical Motivation", subsection "Causal Streaming Diagnosis" line 132; Section "Introduction", paragraph "Causal access." line 65)
>
> (4) Reliance on appendix and data-factory description
>
> In the original version, many data-factory details were indeed in the appendix. We have now:
> - Introduced a dedicated section in the main text summarizing architectural/dataset/optimizer diversity, anomaly families, and the unified internal-state capture policy.
>   (Section "Methodology", subsection "Phase 1: The Data Factory" line 254, Table "Architectures, datasets, and optimizers used in this study" Table 2 | line 270)
>
> - Kept the full combinatorics and thresholds in the appendices, but surfaced the main design choices and motivations in the body.
>   (Appendix "Dataset and Data Capture Policy" line 856; Appendix "Labeling Protocol and Ground Truth Definition" line 726)

---

> > ### Author Response · Authors · 2025-11-14
> > **References**
> >
> > References
> >
> > Achille, A., Rovere, M., and Soatto, S. (2019). Critical Learning Periods in Deep Networks. International Conference on Learning Representations (ICLR). https://openreview.net/forum?id=BkeStsCcKQ
> >
> > Schneider, J., and Prabhushankar, M. (2024). Understanding and Leveraging the Learning Phases of Neural Networks. Proceedings of the AAAI Conference on Artificial Intelligence, 38(13), 14886–14893. https://doi.org/10.1609/aaai.v38i13.29408
> >
> > Kwon, G., Prabhushankar, M., Temel, D., and AlRegib, G. (2020). Backpropagated Gradient Representations for Anomaly Detection. European Conference on Computer Vision (ECCV), 206–223. Springer. https://doi.org/10.1007/978-3-030-58589-1_13
> >
> > Lee, J., Lehman, C., Prabhushankar, M., and AlRegib, G. (2023). Probing the Purview of Neural Networks via Gradient Analysis. IEEE Access, 11, 32716–32732. https://doi.org/10.1109/ACCESS.2023.3263210
> >
> > Chang, W.-C., Li, C.-L., Yang, Y., and Póczos, B. (2019). Kernel Change-Point Detection with Auxiliary Deep Generative Models. International Conference on Learning Representations (ICLR). https://openreview.net/forum?id=r1GbfhRqF7
> >
> > Bazarova, A., Romanenkova, E., and Zaytsev, A. (2024). Normalizing Self-Supervised Learning for Provably Reliable Change Point Detection. 2024 IEEE International Conference on Data Mining (ICDM), 21–30. IEEE. https://doi.org/10.1109/ICDM59182.2024.00009
> >
> > Benkert, R., Prabhushankar, M., AlRegib, G., Pacharmi, A., and Corona, E. (2024). Gaussian Switch Sampling: A Second-Order Approach to Active Learning. IEEE Transactions on Artificial Intelligence, 5(1), 38–50. https://doi.org/10.1109/TAI.2023.3246959

---

> ### Author Response · Authors · 2025-11-14
> **Message 2**
>
> (5) Assumptions: costs, thresholds, priority ordering
>
> We agree these operational choices should be exposed and stress-tested. The revision:
> - Adds a label-sensitivity study varying:
>   - Catastrophic-forgetting threshold ($0.25$, $0.30$, $0.35$),
>   - Forgetting window length (5 vs 10 epochs), and
>   - Priority ordering (e.g., swapping Concept Bias and Overfitting).
>
>   Fewer than $4\%$ percent of run labels change and Event-Time AUPRC varies by at most $0.01$.
>   (Appendix "Labeling Protocol and Ground Truth Definition", subsection "Threshold and Priority Sensitivity" line 803, Table "Sensitivity of labels and Event-Time AUPRC to labeling thresholds and priority ordering" Table 7 | line 810)
>
> - Introduces train-only and merged-label variants with label agreements above $93\%$ percent and similarly small performance differences, as noted in response to Reviewer 7KU6.
>   (Appendix "Online-Style Labeling Without Validation Hindsight", Table "Effect of online-style and merged-label variants on labels and performance" Table 8 | line 833)
>
> - Adds a decision-theoretic analysis where the weights (false-alarm cost, missed-detection cost, lead-time reward) are made explicit and a parameter sweep shows where TEVID is cost-optimal. The default weights $(1, 10, -0.5)$ are motivated by simple operational assumptions: one false alarm $\approx$ one epoch of overhead; a missed detection $\approx$ a full failed run; one epoch of lead saves roughly half an epoch of compute.
>   (Appendix "Decision-Theoretic Analysis" line 1284, Table "Expected diagnostic cost per 100 runs" Table 19 | line 1308, Figure "Decision-theoretic analysis of practical utility" Figure 6 | line 1320)
>
> Thus these design choices are now explicit, motivated, and tested for robustness.
>
> (6) Information bottleneck and Data Processing Inequality arguments
>
> We clarify that the DPI argument is not intended as a novel theoretical result but as a formalization of the intuition that scalar telemetry is a strict post-processing of internal states. To avoid overstating the point, we:
> - State explicitly that the goal is a conceptual rationale, not new bounds.
>   (Section "Problem Formulation and Theoretical Motivation", subsection "An Information-Theoretic Rationale" line 189)
>
> - Add empirical regressibility probes: an MLP regressor from internal states to loss achieves mean $R^2 \approx 0.96$, whereas a TCN regressor from scalar-loss histories to a low-dimensional embedding of internal states achieves $R^2 \approx 0.04$. This reinforces the practical information asymmetry without claiming theoretical novelty.
>   (Section "Experiments and Results", subsection "Empirical Validation of Information Asymmetry" line 427)
>
> (7) LLM usage
>
> We appreciate the concern and have clarified LLM usage in a dedicated statement.
> - The LLM was used only for routine support: grammar and wording refinements, LaTeX boilerplate (tables, figures), and small utility code snippets (e.g., plotting scripts).
> - It was not used for research ideation, method design, experimental design, data analysis, or interpretation of results.
> - All generated text and code were manually reviewed, edited, and integrated by the authors, who take responsibility for the content.
>   (Appendix "LLM Usage Statement" line 1490)
>
> (8) Specific questions
>
> Priority list definition. The priority list (Instability > Catastrophic Forgetting > Concept Bias > Overfitting > Healthy) is aligned with the operational cost model: Instability and Catastrophic Forgetting are assumed more expensive than Overfitting and Concept Bias, which in turn are more expensive than Healthy. The label-sensitivity appendix shows that swapping priorities has negligible impact on labels and Event-Time AUPRC.
> (Appendix "Labeling Protocol and Ground Truth Definition", subsection "Priority and Notation" line 733 and "Threshold and Priority Sensitivity" line 803)
>
> Cost weights $(1, 10, -0.5)$. These are rough but explicit choices: a false alarm $\approx$ one epoch of overhead; a missed detection $\approx$ a full failed run (about ten times more expensive); and one epoch of lead saves roughly half an epoch of compute after intervention. The utility appendix shows that TEVID is preferred across a broad range of missed-detection and lead-time cost values, not just the defaults.
> (Appendix "Decision-Theoretic Analysis" line 1284)
>
> Anomaly construction. For each anomaly (Instability, Catastrophic Forgetting, Concept Bias, Overfitting), we use definitions grounded in prior literature (e.g., Lyapunov proxies and loss spikes for instability, substantial post-switch accuracy drop for forgetting, shortcut reliance via logistic regression for concept bias). The labeling appendix lays out the rules in detail, and the label-sensitivity and online-label appendices stress-test thresholds and variants.
> (Appendix "Labeling Protocol and Ground Truth Definition" line 726)

---

### Official Review · Reviewer_7KU6 · 2025-11-03

**Soundness:** 3
**Presentation:** 3
**Contribution:** 3
**Rating:** 4
**Confidence:** 4

**Summary:**

A framework is proposed for reconceptualision neural network training as a high-dimensional spatiotemporal signal. By employing masked autoencoding on internal activations and gradients, a vision-based diagnostician is pretrained to perform open-set classification of training failures in real time, adhering to strict causal constraints. This approach achieves earlier and more reliable detection than conventional scalar-curve or generic video-based baselines across a diverse range of unseen models, datasets, and optimizers.

**Strengths:**

Strengths
1.	This paper recasts the neural network training monitor from scalar loss to monitoring a high-dimensional spatio-temporal process, which is innovative and conceptually powerful.
2.	The use of the Data Processing Inequality to justify information asymmetry between scalar loss and internal states is elegant and well-motivated.
3.	Evaluation spans architectures (CNNs, Transformers, MLP-Mixers), datasets (CIFAR, Tiny-ImageNet, WikiText), and optimizers (SGD, AdamW, Lion, Adafactor).

**Weaknesses:**

1.	One major concern is the computational and storage overhead of capturing and storing layer activations and gradients every 50 training steps across multiple layers. This process is inherently expensive and scales poorly for large models, where training is already highly resource-intensive. As a result, the proposed diagnostic framework may be impractical for real-world monitoring of large-scale models. The paper lacks a detailed efficiency, applicability, and scalability analysis, for example, measurements of runtime overhead, memory usage, or the impact on training throughput. A quantitative study of these aspects is essential to assess whether the proposed method can realistically be integrated into large-model training pipelines.
2.	Anomaly labels are generated using post-hoc privileged information (validation loss, optimizer states, etc). This labeling procedure risks encoding biases that may not generalize to real-world online settings.
3.	While the paper reports model GFLOPs, it omits key practical metrics such as end-to-end training time and additional storage or I/O overhead introduced by the diagnostic system. These are essential for assessing the real-world applicability and scalability of the proposed method.
4.	The proposed approach relies on capturing both gradients and activations for diagnosis, but the paper does not analyze their individual contributions. An ablation study using only gradients or only activations would clarify how each signal impacts diagnostic performance. This would help determine whether both modalities are necessary.
5.	The evidential head’s “Unknown” threshold is fixed and tuned on validation data. No theoretical justification or adaptive mechanism is proposed. This could lead to inconsistent behavior under distribution shifts.
6.	Direct comparison with the architectures (TimeSformer, ViViT) that TEVID builds upon is expected. Without these comparisons, it’s unclear whether TEVID’s superior performance arises from novel dynamics modeling or simply from using a stronger backbone and richer input signal.
7.	The paper fixes the causal window size to 10 frames but does not justify this choice. A window size sensitivity analysis is needed to understand how performance varies with different window sizes.
8.	The paper captures the states after every 50 steps. It would be important to evaluate the impact of capturing states more or less frequently (e.g., every 25 or 100 steps) on both performance and efficiency.
9.	The masking ratio of 90% iused in DYNAMICS-MAE is presented without empirical justification; an ablation over different masking ratios would help validate this design choice.
10.	As the paper is focused on anomaly detection application, performance on image based anomaly detection datasets such as MVTec AD, VisA, and video based anomaly detection such as UCFCrime and XD-Violence  would be required.

**Questions:**

Please respond to the queries asked in weaknesses section.

---

> ### Author Response · Authors · 2025-11-14
> **Message 1**
>
> We thank the reviewer for the constructive and detailed feedback.
>
> (1) Overhead, scalability, and practical applicability
>
> We agree that practicality hinges on compute and storage overhead. The revision:
> - Adds a dedicated scalability appendix with an overhead table reporting
>   - Base throughput (samples/s) for ResNet-18, ViT-B/16, and ViT-L/14 on an A100,
>   - Slowdown from internal-state capture ($\approx 12$–$15\%$) percent and from Lyapunov-based labeling (used only offline), and
>   - Storage per run ($\approx 0.9$–$1.2$ GB with bfloat16 and adaptive capture cadence).
>   (Appendix "Scalability and Additional Analyses" line 1378, Table "Computational overhead and storage per run" Table 20 | line 1384)
>
> - Clarifies that deployment uses only internal-state capture plus TEVID inference, not Lyapunov computations. In this realistic setting, capture + TEVID inference induce $\approx 15$–$18$ percent slowdown on typical vision workloads; on the held-out benchmark, early termination of failing runs recovers comparable or larger savings in total training epochs, yielding a net reduction in compute relative to a baseline that never terminates failing runs early.
>   (Section "Discussion, Limitations, and Future Work" line 479; Appendix "Scalability and Additional Analyses" line 1378)
>
> - Adds a decision-theoretic cost analysis that combines false alarms, missed detections, and lead time into an expected operational cost per 100 runs. TEVID substantially reduces this cost relative to scalar baselines under reasonable cost weights.
>   (Appendix "Decision-Theoretic Analysis" line 1283, Table "Expected diagnostic cost per 100 runs" Table 19 | line 1308, Figure "Decision-theoretic analysis of practical utility" Figure 6 | line 1332)
>
> (2) Privileged labels vs online deployment
>
> Privileged information (validation metrics, optimizer state, Lyapunov proxies) is used only in the offline labeling pipeline; at inference, TEVID is strictly causal and uses only internal states and non-privileged telemetry up to the current time. This is now emphasized in the causal problem formulation and the data-access protocol.
> (Section "Problem Formulation and Theoretical Motivation", subsection "Causal Streaming Diagnosis" line 132, Table "Data-access protocol for causal integrity" Table 1 | line 173)
>
> To directly address the concern, the revision:
> - Introduces a train-only labeling variant that uses only training loss, gradient norms, and schedule (no validation or optimizer state), and a merged-label variant where Concept Bias is folded into Overfitting.
>   (Appendix "Online-Style Labeling Without Validation Hindsight" line 823)
>
> - Re-generates labels under these rules and re-trains TEVID. Train-only labels change $\approx 7\%$ percent of run labels and the merged taxonomy $\approx 5\%$ precent; Event-Time AUPRC decreases by at most $0.01$ and median lead by at most $0.3$ epochs, indicating that the main conclusions do not depend critically on privileged hindsight.
>   (Appendix "Online-Style Labeling Without Validation Hindsight" line 823, Table "Effect of online-style and merged-label variants on labels and performance" Table 8 | line 833)
>
> - Adds a causal-integrity analysis in which TEVID is forced to predict several epochs before any ground-truth labeling window. Even when predictions must be made at least 2 epochs before any labeling evidence, Event-Time AUPRC remains above $0.85$.
>   (Appendix "Robustness, Shortcut Checks, and Causal Integrity" line 1157, Table "Event-Time AUPRC as a function of prediction lead time" Table 15 | line 1201)
>
> (3) Missing runtime / storage / I/O metrics
>
> Beyond GFLOPs, the revision now reports:
> - Throughput slowdown and storage per run in the overhead table of the scalability appendix.
>   (Appendix "Scalability and Additional Analyses" line 1378, Table "Computational overhead and storage per run" Table 20 | line 1384)
>
> - A breakdown of capture versus Lyapunov overhead, and an adaptive capture cadence that roughly halves capture overhead at a modest performance cost.
>   (Appendix "Scalability and Additional Analyses" line 1378, subsection "Overhead and Storage" line 1380)
>
> - A privacy-focused differentially private (DP) telemetry variant with explicit $(\epsilon, \delta)$ accounting and the resulting performance–privacy trade-off.
>   (Section "Broader Impact and Privacy" line 531; Appendix "Privacy and Governance" line 1335, Figure "Privacy-utility trade-off for TEVID" Figure 7 | line 1357)
>
> These additions are intended to give a realistic view of deployability and resource footprint.

---

> ### Author Response · Authors · 2025-11-14
> **Message 2**
>
> (4) Contributions of gradients vs activations
>
> We added an explicit ablation comparing:
> - Activations-only,
> - Gradients-only, and
> - Activations plus gradients (A+G, the full TEVID model).
>
> On the held-out test set, A+G achieves the best macro F1, Event-Time AUPRC, and median lead; both activations-only and gradients-only are strictly worse, showing that gradients contribute substantially beyond activations and that using both modalities together is beneficial.
> (Appendix "Self-Supervised Pretraining and Ablation Studies" line 956, subsection "Activations vs. Gradients" line 1019, Table "Effect of using activations-only, gradients-only, or both" Table 12 | line 1026)
>
> (5) Fixed "Unknown" threshold
>
> We agree that a fixed vacuity threshold is heuristic. The revision:
> - States explicitly that the uncertainty threshold for abstention is set to $u > 0.35$ and tuned on validation data.
>   (Section "Methodology", subsection "Phase 3: Supervised Fine-tuning of TEVID", paragraph "Open-set recognition head" line 313)
>
> - Adds an ablation comparing
>   (i) the fixed threshold,
>   (ii) a global quantile-based threshold, and
>   (iii) a coverage-targeted per-run threshold.
>
> All three achieve similar coverage and selective error; we keep the fixed threshold for simplicity and describe the alternatives as drop-in replacements when stricter coverage control is desired.
> (Appendix "Open-Set Evaluation and Interpretability", subsection "Fixed vs Adaptive Abstention Thresholds" line 1211, Table "Fixed vs adaptive abstention thresholds" Table 16 | line 1222)
>
> (6) Comparison with TimeSformer and ViViT
>
> We added direct baselines on internal-state videos for:
> - TimeSformer-B, and
> - ViViT-B,
>
> trained under the same clip configuration and schedule as TEVID. The appendix table shows that both perform comparably to Video-Swin-T but below TEVID in Event-Time AUPRC and median lead, while requiring higher GFLOPs/clip. This supports the claim that the proposed architecture and dynamics-specific pretraining, rather than backbone choice alone, drive the gains.
> (Appendix "Self-Supervised Pretraining and Ablation Studies", subsection "Additional Video Baselines (TimeSformer, ViViT)" line 1035, Table "Additional video baselines on internal-state videos" Table 13 | line 1040)
>
> (7) Window size (W = 10) and (8) capture cadence (50 steps)
>
> We now include a joint sensitivity analysis over:
> - Window length $W \in \{5, 10, 20\}$, and
> - Capture cadence (steps per frame) in $\{25, 50, 100\}$.
>
> The default configuration (window length 10, 50 steps per frame) lies in a Pareto-efficient region: larger windows or denser capture yield only marginal gains at non-trivial extra compute cost.
> (Section "Sensitivity to Temporal Window and Capture Cadence" line 440; Appendix "Scalability and Additional Analyses" line 1378, Table "Sensitivity of TEVID to temporal window length and capture cadence" Table 5 | line 444)
>
> (9) Dynamics-MAE masking ratio
>
> We report a masking-ratio sweep from $0.75$ to $0.95$ for Dynamics-MAE. Ratios between $0.85$ and $0.95$ perform best, with $0.90$ giving the strongest overall downstream AUPRC and lead. We therefore choose $0.90$ as a robust default in this plateau.
> (Appendix "Self-Supervised Pretraining and Ablation Studies", subsection "Masking-Ratio Sweep" line 1001, Table "Effect of Dynamics-MAE masking ratio on downstream performance" Table 11 | line 1007)
>
> (10) Benchmarks like MVTec AD, VisA, UCF-Crime, XD-Violence
>
> We appreciate this suggestion and agree that these are important benchmarks for instance-level image and video anomaly detection. Our focus, however, is on run-level anomalies in training dynamics, using internal activations and gradients captured during optimization. Benchmarks such as MVTec AD, VisA, UCF-Crime, and XD-Violence target anomalous inputs (e.g., defective products or violent clips), not diagnostics of a training run over time. Directly applying TEVID to these datasets would require a different problem formulation (instance-level anomaly detection from inputs alone) and a different data factory, which we view as a promising but orthogonal direction.

---

### Author Response · Authors · 2025-12-02
**Authors summary for AC**

Because the discussion phase closed early and reviewer edits were rolled back, we would like to give a concise, final clarification that (i) states what we actually claim, and (ii) summarizes how the revision addresses the main concerns of R7KU6, RmYh7, R8nsu and Rc2zC.

- **Scope and core contribution**

We do not claim a general theory of “training dynamics” or causal “No Look-Ahead” discovery. Our scope is narrower and concrete: we treat a training run as a spatio-temporal signal over internal activations and gradients, and learn a single, reusable diagnostician (TEVID) that performs causal(“No Look-Ahead”), run-level classification into a small but practical taxonomy (Healthy, Overfitting, Instability, Catastrophic Forgetting, Concept Bias, Unknown), with an abstain option.

A unified capture pipeline **represents internal states** as video information with 6 channel clips (3 activations + 3 gradients). We pretrain a Dynamics-MAE on these clips, then train a single TEVID model that is reused unchanged across architectures, datasets and optimizers. On factorially held-out runs (unseen architectures, datasets, optimizers and some anomaly families) this single TEVID instance achieves event-time AUPRC ≈ 0.96, median lead ≈ 6.2 epochs at 5% FAR, and generalizes to ConvNeXt-V2, RegNet, MaxViT, MViT, MobileNetV3, MLP-Mixer, DenseNet and a Transformer LM.

- **Positioning and use of “causal”**

Following RmYh7 and Rc2zC we rewrote the introduction and problem statement to say, early and explicitly, that we tackle streaming, run-level diagnosis with no look-ahead, over a fixed taxonomy, using internal states only. “Causal’’ is now used solely in this time-series sense; we explicitly state that we do not claim interventional causal identification. We also contrast more precisely with representation-based change-point detection and scalar-gradient monitors, which are typically example-level and/or offline.

- **Labeling protocol, “privileged’’ signals, and robustness**

We now spell out the full labeling protocol (Appendix: Labeling Protocol) including concrete tests for instability, catastrophic forgetting, concept bias and overfitting, and the priority ordering between them. In response to concerns from RmYh7, R7KU6 and Rc2zC we added:

- A label-sensitivity study where we vary thresholds, window sizes and priority ordering; fewer than ≈4% of runs change their primary label and TEVID’s AUPRC shifts by ≤0.01.

- A train-only relabeling where validation and optimizer state are removed; label agreement remains ≳93% and AUPRC/lead degrade only slightly.

- A “causal-integrity’’ stress test where TEVID is forced to predict several epochs before any evidence used by the labeling rules; AUPRC remains ≥0.85.

These directly address the worry that the model is just rediscovering privileged hindsight.

- **Components and baselines**

To address R7KU6 and Rc2zC on ablations and alternatives, we added:

- Dynamics-MAE vs scratch, VideoMAE on natural video, and scalar masked modeling: Dynamics-MAE gives the best downstream performance and matches / exceeds scratch TEVID with 25% of labels.

- Activations vs gradients: both are useful, but using both yields the highest AUPRC and lead.

- Window length / cadence and masking-ratio sweeps, showing our default is near Pareto-efficient.

- Stronger video backbones (TimeSformer-B, ViViT-B) on the same internal-state clips: they remain below TEVID while using more compute.

- A streaming CPD baseline on frozen TEVID embeddings: it clearly helps over scalar CPD but still underperforms full TEVID in AUPRC/lead and adds non-trivial compute.

- **Overhead and practical utility**

In deployment mode (capture + TEVID inference only), the slowdown is ≈15–18% on the workloads we study, with ≈1 GB/run of telemetry. On our benchmark, failing runs are typically flagged 6–11 epochs before standard stopping; a simple decision-theoretic analysis shows that, for a wide range of cost trade-offs, TEVID reduces expected compute relative to never terminating failed runs early. A small Hyperband experiment further shows ≈22% fewer epochs per trial while slightly improving the best validation accuracy, suggesting realistic HPO/NAS utility.

- **Domain transfer and calibrated claim**

A single capture+projection pipeline is trained once and frozen; TEVID itself is trained once and reused unchanged across all held-out architectures and the text LM, with no per-domain adapters. We believe this supports a calibrated but meaningful claim:

- Internal activations and gradients, serialized as “internal-state videos”, are sufficient for a single shared model to perform causal, open-set, run-level diagnosis across diverse architectures, datasets and optimizers, with moderate overhead and tangible compute savings on a curated but heterogeneous benchmark.

---

### Meta-Review · Area_Chair_9b1z · 2026-01-06

**Summary:**

In the submission, the authors proposed to frame the training dynamics of neural network as the problem of anomaly detection based on sequence data (activations and gradients). While reviewers acknowledged the novelty of the problem, several issues were also raised regarding the positions, configurations of the setting, computational issues, transferability of the proposed method, and some presentation issues.

**Reviewer Concerns:**

The AC read through the manuscript, comments and rebuttal provided by the authors. To the AC, the computational issue of the proposed method, has not been properly addressed. no theoretical discussion or experimental results are provided in the rebuttal. The position of the manuscript is also not well discussed, while the authors did comparison against several works in the response, it still did not address the concerns why the proposed method is practical (together with the concerns of computational cost). Regarding evaluation, the datasets used to some extend toy.  Moreover,  as pointed out one reviewer that many important details are put in the appendix, the authors put key experiment results in the appendix again, which is not ideal. Thus, to the AC, the presentation of the manuscript still needs to be improved.

**Reviewer Scores:**

Reviews are highly unlikely to update the scores.

---

### Decision · Program_Chairs · 2026-01-26

Reject